# A MECHANISTICALLY INTERPRETABLE NEURAL NETWORK FOR REGULATORY GENOMICS

## ABSTRACT

Deep neural networks excel in mapping genomic DNA sequences to associated readouts (e.g., protein–DNA binding). Beyond prediction, the goal of these networks is to reveal to scientists the underlying motifs (and their syntax) which drive genome regulation. Traditional methods that extract motifs from convolutional filters suffer from the uninterpretable dispersion of information across filters and layers. Other methods which rely on importance scores can be unstable and unreliable. Instead, we designed a novel *mechanistically interpretable* architecture for regulatory genomics, where motifs and their syntax are directly encoded and readable from the learned weights and activations. We provide theoretical and empirical evidence of our architecture's full expressivity, while still being highly interpretable. Through several experiments, we show that our architecture excels in *de novo* motif discovery and motif instance calling, is robust to variable sequence contexts, and enables *fully interpretable* generation of novel functional sequences.

## 1 INTRODUCTION

Transcription factors (TFs) are proteins that regulate genes by recognizing and binding to specific short DNA sequence patterns—or "motifs"—in the genome (Lambert et al., 2018; Siggers & Gordân, 2014). High-throughput experiments measure regulatory activity—such as protein–DNA binding or associated readouts—across the genome (Consortium, 2012). In general, the regulatory function of a DNA sequence is defined by the combination of motifs in that sequence, and these combinations have a soft syntax (i.e., density, spacing, orientation, and co-binders). These syntactical rules also depend on surrounding context and cell type. Although challenging, understanding the motifs (and their syntax) which regulate the genome is crucial for many scientific and medical tasks, such as disease diagnosis and design of novel therapies (e.g., CRISPR). There is thus a great need for techniques which perform *de novo* motif discovery (i.e., extracting motifs and syntax from these experiments).

Recently, *de novo* motif discovery has been chiefly performed by *sequence-to-function deep neural networks* (DNNs). These DNNs take a DNA sequence as an input, and predict a functional label measured by a biological experiment (e.g., a binary label denoting if a TF bound to that sequence). These expressive models have achieved state-of-the-art performance in mapping DNA sequences to regulatory labels, thus proving that they accurately learn underlying motifs and their syntax (Kelley et al., 2016; Žiga Avsec et al., 2021b; Linder et al., 2023). The common goal of these DNNs is to ultimately reveal these learned motifs and syntax that underpin genome regulation (Novakovsky et al., 2022a). Notably, recent work has shown that motif discovery from these sequence-to-function DNNs has far surpassed the ability of traditional statistical methods (Tseng, 2022).

There are two main classes of methods that extract motifs from a trained DNN. As these DNNs are almost universally convolution-based in early layers (Novakovsky et al., 2022a), a very common approach is to extract a motif from each first-layer filter (Kelley et al., 2016). This method, however, suffers from the critical limitation that information (including motifs) tends to be *distributed* or *dispersed* across filters and layers, thus there is no guarantee that any single filter will learn a biologically meaningful motif (Figure 2a). The second class of methods relies on importance scores, which attempt to measure the contribution of individual DNA bases to the output prediction, with the assumption that bases in motifs have elevated importance. By integrating throughout the whole DNN, importance scores bypass the problem of distributed information. Unfortunately, importance scores are known to be highly unstable and unreliable approximations of a model's decision making

(Ghorbani et al., 2017; Alvarez-Melis & Jaakkola, 2018), and in genomic DNNs, importance scores typically only show noisy and fragile motif instances (Supplementary Figure S1) (Tseng et al., 2020). In practice, to improve robustness, motif discovery via importance scores requires complex pipelines composed of many computationally expensive steps, which tend to be delicate and require constant human intervention (Novakovsky et al., 2022a; Shrikumar et al., 2018).

In this work, we propose a new method of recovering motifs from a sequence-to-function DNN, based on *mechanistic interpretability*. Mechanistic interpretability (MI) has recently emerged as a key research direction to explain complex models (Bereska & Gavves, 2024). Our method, **Analysis of Regulatory Genomics via Mechanistically Interpretable Neural Networks (ARGMINN)**, enables motifs *and* their syntax to be directly readable from the network architecture, without compromising expressivity or relying on complex *post hoc* pipelines. In Section 3, we formally describe the ARGMINN architecture, including several novel architectural contributions including: **1)** a regularizer designed to ensure that the first layer's convolutional-filter weights directly encode a non-redundant set of relevant motifs; and **2)** a modified attention mechanism which reveals motif instances and their syntax in any query sequence with a single forward pass.

In Section 4, we show experimental results which demonstrate ARGMINN's interpretability and its main contributions to regulatory genomics, including: **1)** superior motif discovery *and* motif-instance/syntax analysis compared to existing approaches for *de novo* motif discovery; **2)** robustness against natural or adversarial sequence modifications; and **3)** the novel ability for *fully interpretable* sequence generation. In Section 5, we provide theoretical results on ARGMINN's expressivity, showing that it can learn any motifs and syntax, whereas previous MI architectures cannot.

## 2 RELATED WORK

Practically all prevalent genomic DNN architectures have convolutional filters as the first layer (Kelley et al., 2016; Žiga Avsec et al., 2021b; Linder et al., 2023). To recover motifs, most works directly visualize the filter weights or average subsequences which highly activate each filter (Alipanahi et al., 2015; Kelley et al., 2016). Although this has shown some limited success, these methods assume that each filter learns one motif, and each motif is learned by one filter. This is generally not true because—without special constraints—motifs are typically learned in a *distributed* fashion, where each motif is learned across many filters and layers (Figure 2a).

As a result, more sophisticated *post hoc* pipelines were developed to extract motifs from trained DNNs. These pipelines integrate over the entire DNN to compute importance scores across the dataset (e.g., via integrated gradients or DeepLIFTShap (Sundararajan et al., 2017; Shrikumar et al., 2017)), resulting in an importance score at each position for each sequence. These scores are then segmented into high-importance regions as putative motif instances. Due to the noisiness of importance scores, however, these instances must be clustered into clean motifs by tools like MoDISco (Shrikumar et al., 2018). Each step of this pipeline is computationally expensive, and for most datasets, the time required to recover motifs is typically over an order of magnitude longer than the time needed to train the model. Furthermore, these pipelines heavily rely on importance scores from a black-box model, which can be extremely fragile, as importance scores frequently fail to reveal a model's true decision-making process (Ghorbani et al., 2017; Kindermans et al., 2017; Alvarez-Melis & Jaakkola, 2018; Tseng et al., 2020).

Within the field of explainable AI, there has been some burgeoning work exploring MI, where the patterns and rules learned by a DNN are reflected in its physical computation (Bereska & Gavves, 2024). In particular, our work is a type of *intrinsic* MI, where the DNN's architecture (e.g., weights and activations) directly encode learning (Liu et al., 2023; Barbiero et al., 2023; Kasioumis et al., 2021). This is typically done by increasing sparsity, modularity, and the proportion of monosemantic neurons (i.e., neurons which learn a single concept). DNNs may also improve their intrinsic MI by learning logical rules in a more structured way (Riegel et al., 2020; Friedman et al., 2023).

Despite these promising works, constructing intrinsically MI architectures for very general problems remains difficult. However, focusing on more constrained predictive tasks (e.g., motif discovery) makes MI more feasible, allowing us to restrict computation and the solution space. ARGMINN uses similar principles as other intrinsic MI works (e.g., sparsity, modularity, monosemanticity, and explicit logic), but in a biologically grounded manner, making it both highly interpretable and expressive.

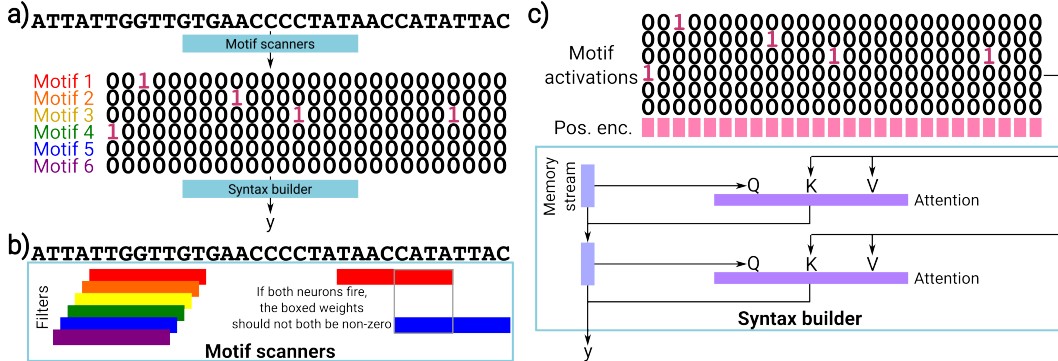

Figure 1: Schematic of the ARGMINN architecture. **a)** The motif-scanner module produces activations denoting which motifs were found at each position, where activation magnitude reflects match strength. The activations are passed to the syntax builder, which learns higher-order logic between motif instances for the final prediction. **b)** The motif-scanner module is a single convolutional layer which learns all motifs *de novo*. Regularization ensures that each filter learns one motif (and *vice versa*), penalizing different filters from activating based on the same underlying subsequences. **c)** The syntax builder is a series of uniquely designed attention layers. An explicit memory stream tracks the model's state. Each attention layer derives a single query vector from the memory stream, and key/value vectors from the original activations, to update the memory stream.

Recently, the ExplaiNN architecture also aimed to interpretably learn motifs from DNA sequences (Novakovsky et al., 2022b). ExplaiNN consists of a set of single-filter convolutional networks. Each network ideally learns a distinct motif, and outputs a scalar summarizing binding strength over the input. The final output is a learned linear combination of these scalars. In later sections, we will show that ARGMINN surpasses ExplaiNN in both interpretability and expressivity.

Our work also relates to concept bottleneck models (CBMs) (Koh et al., 2020; Ghorbani et al., 2019; Kim et al., 2017), which force decisions to be based on interpretable concepts from an intermediate layer. While CBMs can provide meaningful explanations, they require pre-defined concepts and concept-labeled inputs, which are labor-intensive to obtain. In our work, ARGMINN can be viewed as a type of CBM, but the concepts are motifs, and *the concepts are learned entirely from the data*, thus overcoming the typical limitations of CBMs. Furthermore, ARGMINN elucidates syntax between motifs—rules relating to *positioning* between concepts—in addition to the motifs themselves.

## 3 ARGMINN ARCHITECTURE

We propose a mechanistically interpretable DNN architecture designed to 1) accurately predict regulatory function (e.g., protein binding) from DNA sequence; 2) reveal crucial motifs responsible for function across the dataset; and 3) reveal motif instances and their syntax in any given query sequence. Importantly, 2) and 3) are directly encoded in the model's weights and activations.

Our architecture consists of two modules trained end to end (Figure 1a). The *motif-scanner* module is a single convolutional layer which identifies motifs from the input sequence. We developed a novel regularizer for the filters so that the filter weights directly encode non-redundant motifs, thereby accomplishing goal 2) above. The convolutional activations are passed to the second module—the *syntax builder*—consisting of several layers of modified attention that assemble the syntax and logic between motifs to produce a final prediction. The syntax builder is designed such that the activations and attention scores immediately reveal motif instances and syntax, thus achieving goal 3).

### 3.1 MODULE 1: MOTIF SCANNERS

The "motif scanners" are a set of $n_f$ convolutional filters of width $w$, which are scanned across the input sequence to produce a set of activations (Figure 1b). Thus, a filter's activation is maximized when it scans over a one-hot-encoded sequence that exactly matches the encoded motif (motifs are learned *de novo*). The activations are then thresholded by a ReLU function. The additive

bias parameter of the filters and the ReLU allow the network to selectively cut off weak matches, thereby producing more sparse activations. For a 1-hot encoded DNA sequence $S \in \{0,1\}^{\ell \times 4}$, the convolutional weights $W \in \mathbb{R}^{n_f \times w \times 4}$ (and bias $b \in \mathbb{R}^{n_f}$) yield activations $A \in \mathbb{R}_+^{(\ell-w+1) \times n_f}$:

$$A = \text{ReLU}(\text{Conv}(S, W, b)). \tag{1}$$

Importantly, the filters are regularized so that each filter learns one distinct motif, and each motif is learned by one filter. This allows motifs to be directly read from the filter weights after training. We propose a novel secondary objective which penalizes different filters from activating on overlapping sequences. This is combined with a simple L1-penalty on the filter weights themselves to induce sparse filters that directly reveal *distinct, non-redundant* motifs.

At each position $i$ in sequence $S$, each filter aggregates values $S[i, i+w]$. Let $a = \text{argmax}\{A_i\} \in \{0, \ldots, n_f - 1\}$ be the index of the maximally activated filter at $i$. For all other filters $b \neq a$, if filters $a$ and $b$ both achieve non-zero activation in the same neighborhood of $S$, then they should not both have non-zero weights in the overlapping region (i.e., they should not be attending to the same part of the sequence). In other words, every position of $S$ is "protected": at most one filter can activate while attending to any position. If another filter is activated nearby, then its weights should not be attending to the protected part of the sequence. Our *filter-overlap* regularization then can be defined as the following loss function:

$$\mathcal{L}_o(A, W) = \sum_{i=0}^{\ell-w} \sum_{b \neq \text{argmax}\{A_i\}} \sum_{j=i-(w-1)}^{i+(w-1)} \Big[ A_{j,b}$$
$$\|W_{\text{argmax}\{A_i\}}[\max\{0, j-i\}, w-1-\max\{0, i-j\}]\|_1$$
$$\|W_b[\max\{0, i-j\}, w-1-\max\{0, j-i\}]\|_1 \Big]. \tag{2}$$

At each position $i$, we compare the filter weights of the maximally activated filter $W_a$ with the weights of all other filters ($W_b$), at every possible overlapping window $j$. We penalize the L1 norm of the overlapping weights, multiplied by the activation of filter $b$ (i.e., if $W_b$ is not activated, there is no penalty). Importantly, this is a *soft* regularization which the model can choose to ignore if needed for performance. This regularization helps prevent: 1) two filters learning the same motif (or the same part of the same motif); and 2) two filters learning a motif in an interleaved fashion. However, our regularization still allows for a long motif to be learned by two filters, split somewhere down the middle (or similarly, two half-sites directly next to each other, each learned by one filter).

In practice, this loss is computed efficiently by caching all possible windows of weight sums (for each value of $j$) once, and at each $i$ scaling the window products with the activation of $W_b$.

## 3.2 MODULE 2: SYNTAX BUILDER

After the motif-scanner module, positional encodings $P$ are concatenated to the activations $A$. The second module of ARGMINN is the syntax builder, which consists of $n_L$ layers of a custom attention mechanism that learns syntax between motifs (Figure 1c). In contrast to typical attention, our modified attention layer has an explicit "memory stream" $m_l$ which is updated after each layer (inspired by Friedman et al. (2023)). Each layer derives a *single* query from $m_l$, resulting in a linear vector of attention scores rather than a quadratic matrix (this improves both interpretability and efficiency). Importantly, every layer derives key and value vectors *directly* from the original "tokens" (i.e., $A\|P$). Each attention layer can be described as follows:

$$q_l := W_{Q,l} m_{l-1}, \quad K_l := W_{K,l}[A\|P], \quad V_l := W_{V,l}[A\|P]$$
$$a_l := \frac{K_l q_l}{\sqrt{d_{q_l}}}, \quad m_l := m_{l-1} + \text{MLP}(a_l V_l), \tag{3}$$

where $m_0$ is initialized as a vector of ones, and $d_{q_l}$ is the dimension of the query vector. We also include $n_h$ attention heads, but do not show the reshaping operations above for clarity.

Each layer can attend to multiple motifs (due to the multiple attention heads), and successive layers allow the model to capture *interactions* between motifs (e.g., with $k$ layers, the model can reason about $k$th-order interactions between motifs). All together, our final loss function becomes:

$$\mathcal{L}(S, A, W) = \mathcal{L}_{pred}(f(S), y) + \lambda_o \mathcal{L}_o(A, W) + \lambda_l \|W\|_1. \tag{4}$$

## 4 EXPERIMENTAL RESULTS

In this section, we show experimental results demonstrating ARGMINN's advantages in motif discovery, motif instance calling, robustness, and interpretable generation.

### 4.1 IMPROVED MOTIF DISCOVERY

To extract motifs from ARGMINN, we applied the procedure from Kelley et al. (2016) to the first-layer convolutional filters. Specifically, we obtained a motif from each filter by averaging test-set subsequences which highly activate the filter (filters which were never activated in the test set were dropped) (Supplementary Methods C.4).

Over several simulated and real-world experimental datasets (Supplementary Methods C.1), we compared the motifs discovered by ARGMINN to those identified by several other methods: interpreting the filters of a standard CNN, ExplaiNN (Novakovsky et al., 2022b), and importance-score clustering via DeepLIFTShap and MoDISco (Shrikumar et al., 2018). We systematically matched each discovered motif to the closest known relevant motif. For simulated datasets, we matched to ground-truth motifs; for experimental datasets, we matched to the closest relevant human motif. ARGMINN identified *known, biologically relevant motifs*, and compared to other methods, it generally missed the fewest relevant motifs and discovered the fewest redundant motifs (Figure 2a–b, Supplementary Figures S2–S5, Supplementary Table S1). For example, ARGMINN trained on FOXA2 in HepG2 (a pioneer factor) revealed factors in the FOX, HNF4, and CEBP families, all known to co-localize or co-bind with FOXA (Seachrist et al., 2021; Geusz et al., 2021; Liu et al., 2020). Notably, in the experimental datasets, the singular most similar motif (an extremely strict requirement) to an ARGMINN filter is typically a known relevant motif (e.g., on this FOXA dataset, ARGMINN identified 4 motifs whose top match was a relevant motif, whereas the traditional CNN only identified 2, with much weaker similarities). Additionally, ARGMINN's motifs were generally most similar to the ground truth (Figure 2c, Supplementary Table S2), thus highlighting their *quality*. ARGMINN was also the method which consistently identified the fewest extraneous motifs—patterns which do not match any biologically relevant motif (Figure 2d, Supplementary Table S6).

In contrast, other methods significantly underperformed compared to ARGMINN. Even when the standard CNN or ExplaiNN encoded motifs in their filters, ARGMINN's filters were far more similar to the true motifs. ARGMINN also outperformed MoDISco (Shrikumar et al., 2018), which—despite recovering many relevant motifs—consistently identified less-accurate motifs than ARGMINN (Figure 2c). Due to the unreliability and noisiness of the importance scores themselves, as well as the frailty of clustering, MoDISco also found *many* redundancies and extraneous motifs (Figure 2b, 2d).

### 4.2 IMPROVED MOTIF INSTANCE CALLING AND SYNTAX DISCOVERY

With previous DNN-based methods, identifying motif syntax required first learning motifs (e.g., via MoDISco) and then scanning sequences to "call" motif instances. Not only is this computationally expensive, but instance calling by sequence scanning tends to be highly inaccurate (e.g., due to partial hits or missing context which the DNN would have considered).

Instead, ARGMINN reveals motif instances by tracing attention scores from a single forward pass on any sequence (Figure 3a). Since every attention layer derives keys/values directly from the motif-scanner activations, high attention scores directly point to the precise motif instances which the network deemed important for prediction. Specifically, for any input query sequence, we examine the attention scores across all layers/heads on the forward pass. For each high score (e.g., $> 0.9$), we identify the corresponding sequence position. We then check the motif activations at that position, and call a motif instance if a motif/filter has high activation (Supplementary Methods C.4). For example, trained on an experimental dataset of REST binding, ARGMINN directly recovered the

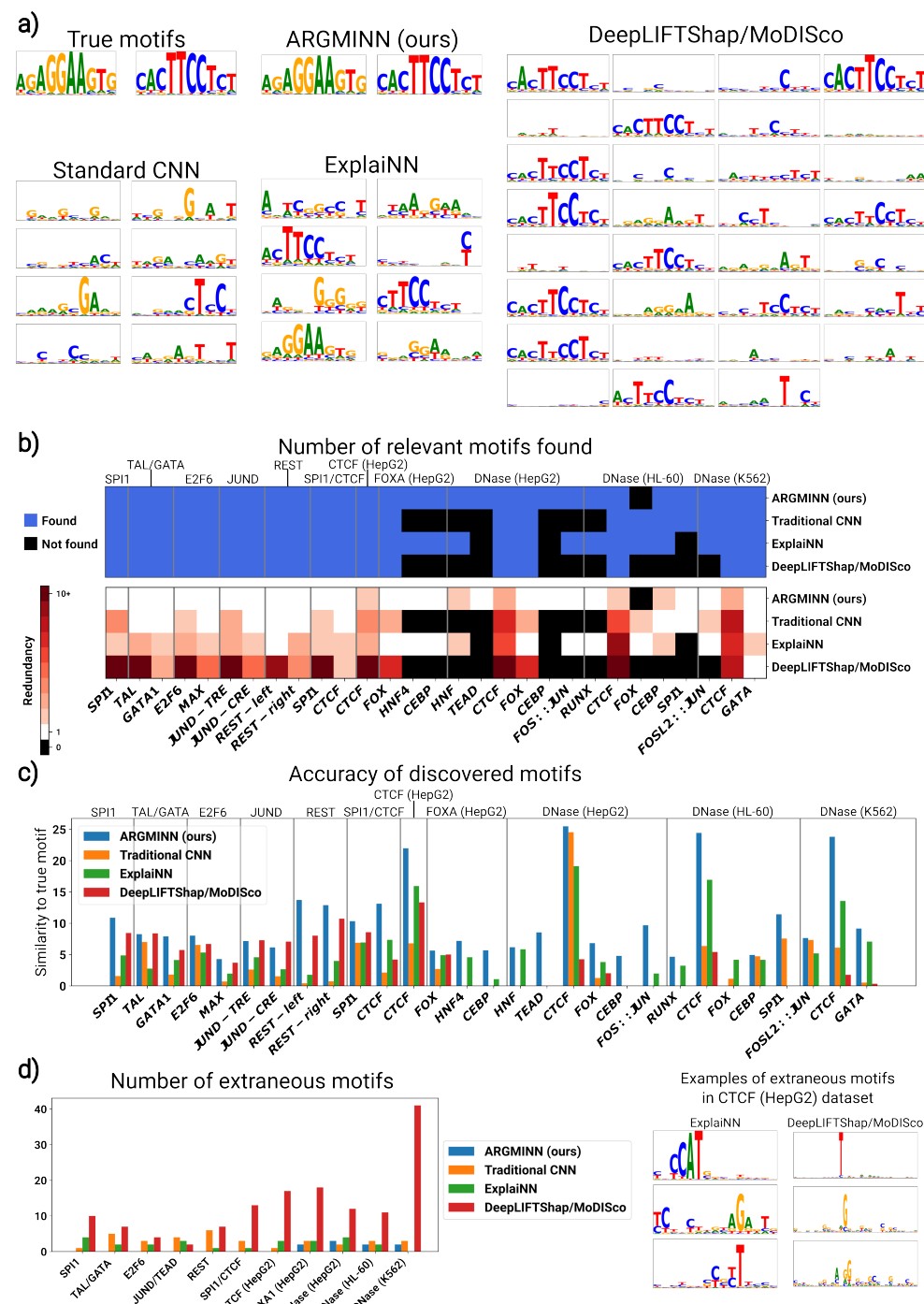

Figure 2: Motif discovery. **a)** Example of SPI1 motifs discovered by ARGMINN, compared to interpreting the first-layer filters of a standard CNN, using ExplaiNN, and by clustering DeepLIFTShap importance scores using MoDISco. Note that MoDISco combines forward and reverse-complement orientations. **b)** For each dataset, we show whether or not each method successfully recovered each relevant motif (above), and the amount of redundancy as the number of times each motif was discovered (below). **c)** To quantify accuracy of the discovered motifs, for each relevant motif we show the maximum similarity to motifs discovered by each method. **d)** For each dataset, we show the number of extraneous motifs—those which do not match any known relevant motif—that each method discovered (left). We show a few examples of such extraneous motifs discovered for the CTCF (HepG2) experimental dataset (right).

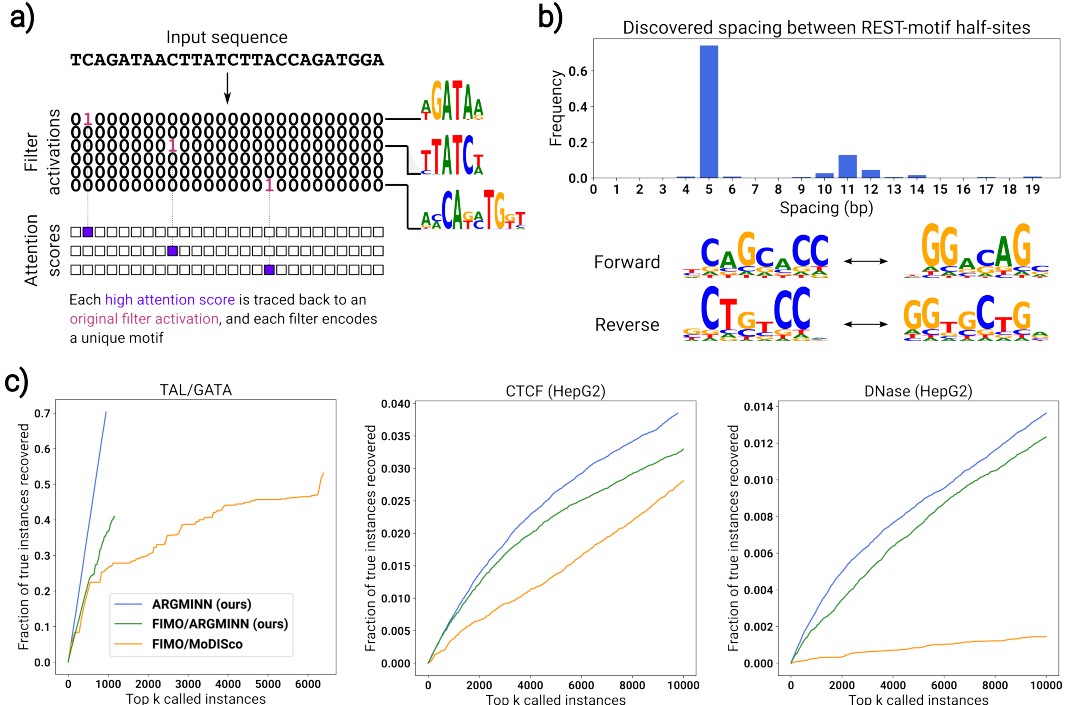

Figure 3: Motif instance calling and syntax discovery. **a)** ARGMINN calls motif instances in any query sequence in only a forward pass. High attention scores in any attention layer trace directly back to original filter activations, which directly map to sequence motifs. **b)** After training on an experimental dataset of REST binding in HepG2, ARGMINN revealed the unique binding syntax of REST in both the forward and reverse-complement orientations, where the half-sites (left and right) bind either adjacently or around 9–14 bp apart. **c)** We compare motif instances discovered by ARGMINN to the traditional approach of using MoDISco to discover motifs and subsequently scanning for them with FIMO. We rank motif instances by confidence (attention score from ARGMINN, or FIMO hit q-value), and compute the fraction of true instances that are covered in a top-$k$ fashion. We also compare to motif instances discovered by scanning for ARGMINN-discovered motifs with FIMO.

unique binding syntax—including spacing preferences—of the two halves of the REST motif in both directions/orientations (Tang et al., 2021) (Figure 3b).

We then quantitatively evaluated motif instances identified by ARGMINN versus a traditional pipeline. Namely, using FIMO (Bailey et al., 2015), we called instances of MoDISco-discovered motifs (Shrikumar et al., 2018) in test sequences. For each method, we ranked instances by confidence and computed the number of ground-truth instances recovered by the top-$k$ called instances. For experimental datasets, we used independently derived binding footprints as ground truth (Vierstra et al., 2020). In general, ARGMINN's motif instances were far more accurate than those found by the baseline (Figure 3c, Supplementary Figure S6, Supplementary Table S4). To gain further intuition, we also compared to motif instances found by using FIMO to scan for ARGMINN-discovered motifs. ARGMINN identified more accurate motif instances than FIMO, even when FIMO was given the same set of ARGMINN motifs. This comparison also shows the direct benefit of using ARGMINN to perform motif instance calling, compared to the traditional method of sequence scanning.

Finally, to further demonstrate that ARGMINN's attention scores match underlying biological signal, we show that the positions of high attention scores in experimental test-set sequences closely track the measured biological strength of protein binding along the sequence (Supplementary Figure S7).

### 4.3 QTL PRIORITIZATION

Next, we evaluated ARGMINN's ability to classify and prioritize a set of causal DNase-sensitivity quantitative trait loci (dsQTLs) from a background set of non-causal dsQTLs. dsQTLs are mutations

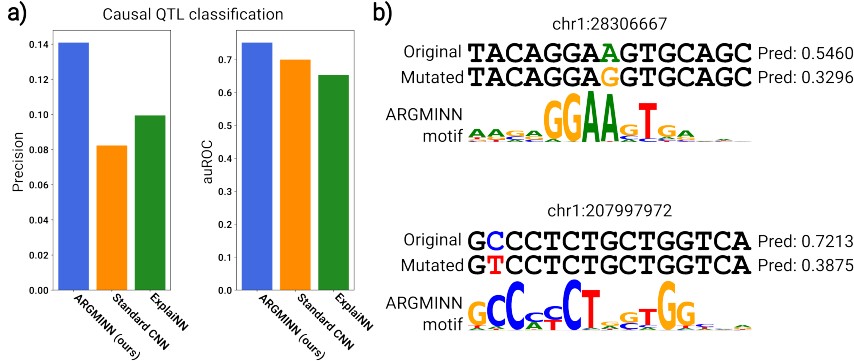

Figure 4: QTL classification. **a)** On a set of known DNase-sensitivity QTLs, we evaluated the ability of ARGMINN to prioritize true causal dsQTLs by quantifying the difference in predictions with and without the dsQTL mutation. **b)** We show two examples of causal dsQTLs, which fall in ARGMINN-discovered motif instances. In both cases, by making the dsQTL mutation, the ARGMINN-discovered binding site is disrupted, leading to a reduced prediction of accessibility.

which change the DNase accessibility of a sequence. A predictive model which makes decisions based on biological signals (and not spurious correlates) should predict a larger change for causal dsQTLs compared to non-causal dsQTLs, which are merely correlated with the causal changes. On a held-out chromosome, we found that compared to other architectures, ARGMINN was capable of classifying/prioritizing the causal dsQTLs much more accurately (Figure 4a). In particular, these causal dsQTLs tended to overlap specific motif instances which ARGMINN used for its prediction of accessibility (Figure 4b). This further shows that ARGMINN makes biologically meaningful predictions on sequence changes, based on interpretable motif biology rather than spurious signals.

### 4.4 ROBUSTNESS OF ARGMINN

Because ARGMINN makes decisions based on biologically meaningful motifs, it is more robust to background variations. On several simulated datasets, we trained ARGMINN (along with a standard CNN and ExplaiNN) with a 50% GC background. We then computed predictive performance on test sequences with GC content ranging from 5% to 95%, thus simulating natural variation in background composition. ARGMINN's performance suffered the least, with a higher and tighter distribution of performance in general (Supplementary Figure S8a).

Furthermore, because traditional CNNs learn filters which do not accurately represent motifs, they are prone to adversarial attacks (Supplementary Methods C.4). On our SPI1 dataset, we trained a CNN to achieve **88%** accuracy. We then easily constructed many sequences containing short substrings which highly activate its filters, but without any instances of the SPI1 motif. On this set of sequences, the standard CNN's accuracy dropped to **55%**. Reflexively, we also easily designed sequences containing the SPI1 motif, but we inserted substrings into the background which strongly deactivate the filters, leading the CNN's accuracy to drop to **48%**. In contrast, because ARGMINN encodes meaningful biological motifs in each filter, it remains robust against such an attack (i.e., one cannot easily identify highly-activating or anti-activating sequences which trick the model into giving the wrong prediction).

Finally, we used Ledidi (Schreiber et al., 2020) to generate sequences which would bind to SPI1. Ledidi is a gradient-based sequence-design method, which optimizes input sequences to maximize the probability of a positive prediction from a given model. Applying Ledidi on the ARGMINN architecture generated sequences with the strongest instances of the true SPI1 motif (Supplementary Figure S8b). The true binding strength of ARGMINN's Ledidi-generated instances was significantly higher than those generated from the standard CNN ($p = 8.95 \times 10^{-18}$) and ExplaiNN ($p = 5.53 \times 10^{-146}$). This is likely due to the fact that ARGMINN is explicitly trained to focus on the relevant motifs, whereas the other models often focus on spurious, non-motif sequence patterns, resulting in ARGMINN's being a more robust oracle for optimization.

4.5 INTERPRETABLE DESIGN OF NOVEL FUNCTIONAL SEQUENCES

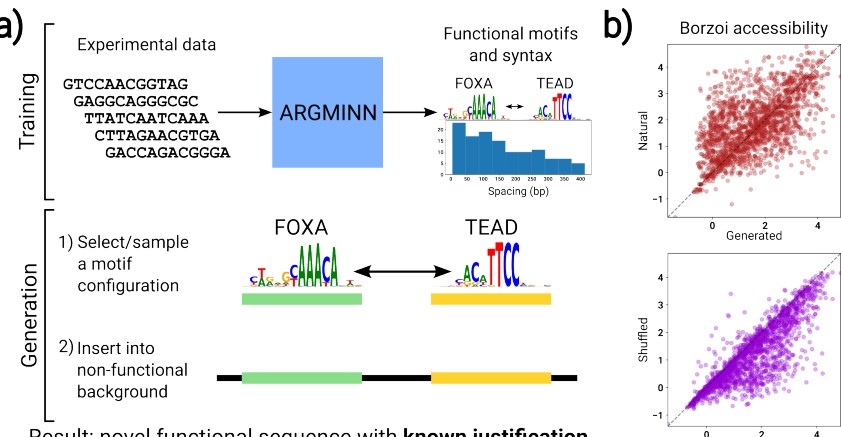

Figure 5: Interpretable sequence design. **a)** After training on experimental data (e.g., HepG2 accessibility), ARGMINN reveals the functional motifs and their binding syntax which induce function. To generate a novel functional sequence, we insert motifs into a non-functional background, following the syntactical rules learned by ARGMINN. In this novel sequence, the mechanistic justification is fully controlled and known. **b)** We interpretably constructed novel HepG2-accessible sequences using ARGMINN and validated their accessibility using Borzoi. We compared the accessibility between generated sequences and natural sequences from the experiment (top), as well as between generated sequences and shuffled backgrounds (bottom).

Upon training, ARGMINN reveals the functional motifs and the syntatical rules for combining them which produce a prediction of binding or accessibility. As such, ARGMINN can be used for *interpretable generation of novel functional sequences*. In contrast with sequence generation using traditional methods (e.g., gradient-based tools like Ledidi, or directed evolution) or non-interpretable generative models (e.g., diffusion models or autoencoders), we may start with ARGMINN interpretations and sample motif configurations to insert into non-functional backgrounds. This yields a generated sequence for which we have a complete understanding on the *reason* why it is biologically active (e.g., it is active because it has the FOXA motif and TEAD motif 50 bp apart) (Figure 5a).

To demonstrate this ability, we trained ARGMINN to predict DNA accessibility in the HepG2 cell type from experimental data. On the test set, we then extracted the functional motifs and the syntactical rules and grammars between them. We sampled motifs and their configurations entirely from ARGMINN's discovered biology, and inserted them into non-functional sequence backgrounds (Supplementary Methods C.4). Using the same model as a predictor of accessibility, we found that the newly generated sequences were predicted to be significantly more accessible than the shuffled non-functional backgrounds ($p = 1.78 \times 10^{-32}$). Furthermore, these generated sequences were predicted to be *just* as accessible as natural sequences identified to be highly accessible by the original biological experiment ($p = 0.75$) (Supplementary Figure S9c). We then repeated our *in silico* validation with a completely independently trained model, Borzoi (Linder et al., 2023). Using the output head of Borzoi which predicts DNA accessibility in HepG2, we passed the same sequences through Borzoi to predict accessibility and confirmed the same trends: the generated sequences were far more accessible than the non-functional backgrounds, and were similar in accessibility to natural sequences identified by the experiment (Figure 5b). We found that this trend was also upheld by two additional independent oracles (Supplementary Figure S9a–b).

## 5 THEORETICAL RESULTS

For a MI architecture, ARGMINN is highly interpretable, yet uniquely retains high expressivity. Here, we show that ARGMINN is capable of learning any possible configuration of motifs—including positional and syntactic constraints—as long as the constraints are definable in first-order logic.

**Theorem 1.** *Every configuration of motifs/subsequences which is definable by a sentence in first-order logic (with positional variables) is recognizable by an ARGMINN classifier.*

Furthermore, ARGMINN is more expressive than previous interpretable architectures for regulatory genomics, such as ExplaiNN (Novakovsky et al., 2022b), which learns an output label from a linear combination of motif strengths:

**Corollary 1.1.** *There exists a configuration of first-order-logic-expressable motifs/subsequences which is* not *recognizable by any ExplaiNN classifier.*

We formally prove Theorem 1 and Corollary 1.1 in Appendix A. Here, we briefly sketch the proof of Theorem 1. Intuitively, ARGMINN's motif-scanner module outputs motif binding strengths by encoding each motif's position-weight matrix (Benos et al., 2002) in a different filter. With expressive positional encodings, the syntax-builder module learns syntax/interactions between motif instances. Each attention head learns one syntactical "rule" (i.e., possible motif combinations), which is built up in complexity over layers (i.e., combinations of $k$ motifs are learned by the $k$th layer).

## 5.1 EXPERIMENTAL FOLLOW-UP TO THEORETICAL RESULTS

To empirically reinforce our theoretical results, we consider the REST binding motif, which consists of two halves which must bind together (with variable spacings) (Figure 3b). Notably, both halves need to bind in the same orientation and order (half sites cannot be mixed/matched). On a simulated REST dataset which explicitly tests these complex requirements, ARGMINN achieved **90.9%** test-set accuracy, whereas ExplaiNN only achieved **73.4%**. This demonstrates that ARGMINN is sufficiently expressive to capture complex grammars and syntax between motifs, which previous architectures could not.

## 6 DISCUSSION

We illustrated ARGMINN's unique ability to reveal genome-regulatory motifs and their syntax directly from its weights and activations—an advantage which is entirely absent from traditional (non-MI) DNNs. We then compared the predictive performance of ARGMINN to standard CNNs and ExplaiNN (Supplementary Table S5). It is generally well known that more interpretable models may suffer slightly in predictive performance (Dosilovic et al., 2018; Arrieta et al., 2020). This is expected, as these models tend to base decisions on human-interpretable concepts (e.g., crucial motifs), instead of spurious signals which can be informative, but are not useful for understanding regulatory genomics (e.g., GC content, exceptionally rare motifs, etc.). Our empirical results fall in line with these expectations. On simulated datasets where we ensured that the only informative signals are motifs, ARGMINN achieved superior performance compared to the baselines. On experimental datasets with many more spurious signals, ARGMINN achieved competitive performance, but did not outperform its non-MI counterparts.

Additionally, to further show the benefit of our filter regularizer, we applied our regularization to a standard CNN. As a result, the CNN's filters also encoded relevant and non-redundant motifs (Supplementary Figure S10). This shows that *our filter-overlap regularizer can be readily applied to standard genomic DNNs* to achieve more interpretable first-layer filters in general. Importantly, however, without the other architectural novelties of ARGMINN (Equation 3), such a DNN would still not reap benefits such as motif-instance and syntax discovery.

Finally, we explored the robustness of ARGMINN to the loss weights (which are hyperparameters) for our regularizers. We found that over many orders of magnitude, ARGMINN's *predictive performance and interpretability both remained robust to the loss weights* (Supplementary Figure S11). This is partially due to the design of these regularizers, which are aimed to *synergize* with predictive performance and the learning of biological motifs, rather than compete with other losses.

Mechanistic interpretability is still a nascent field, and our research pioneers an architecture that enables direct interpretation with minimal *post hoc* analysis. To our knowledge, ARGMINN is the most MI (yet still fully expressive) architecture for genome regulation—a field where *understanding* a model's learned decision rules is equally as critical as its accuracy (Eraslan et al., 2019; Rudin, 2019)—and is one of the few intrinsically MI architectures of its complexity in general. We showed that ARGMINN is expressive enough to accurately predict genome regulation, yet is uniquely constrained so the weights and activations directly encode the decision process *in a human-interpretable way*. Further work in this area can yield major benefits for scientific AI and explainable AI.

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

# A    SUPPLEMENTARY PROOFS

## A.1    PROOF OF THEOREM 1

In this section, we prove that every configuration of motifs (which is definable by a first-order-logical sentence with position-indexed variables) is recognizable by an ARGMINN classifier. We will first show that ARGMINN's motif-scanner module is sufficiently expressive to characterize motif-based protein binding, and then subsequently that the syntax-builder module can recognize arbitrary logical syntax between these binding sites.

**Biophysical assumptions**

We begin with our biophysical assumptions which justify our modeling of motif biology by first-order logic (with position-indexed variables).

1. For binary biological readouts of interest (e.g., protein-binding measured by ChIP-seq, DNA accessibility measured by DNase-seq, etc.), the readout is fully characterized by the binding of proteins, which recognize motifs in the sequence (Stormo & Fields, 1998; Lemon & Tjian, 2000).

2. The free concentration of any particular protein is constant across training and testing conditions.

3. For a given potential binding site (e.g., a motif instance in a DNA sequence), the strength and likelihood of binding of a specific protein (these quantities are related through statistical mechanics) can be sufficiently summarized by a single scalar value (related to the $K_d$, or dissociation constant). Through statistical mechanics, the fraction/probability of binding is $\frac{[\text{TF}]}{K_d + [\text{TF}]}$ (Stormo & Fields, 1998).

4. The $K_d$ value (or a monotonic function of it) can be computed as an independently additive function of individual positions of the binding site. An example of this is the PWM (position weight matrix), which has been shown to be a good approximation of binding mechanics (Benos et al., 2002; Pan & Phan, 2008).

5. Input sequences are a finite length $\ell$.

6. Given real-valued variables $b_{m,p}$ representing the binding strength of motif $m$ at position $p$ in a sequence, the binary biological readout (e.g., binding of a particular protein of interest or accessibility) of the sequence as a whole can be expressed as the following disjunction of statements:

   $$\sigma := \bigvee_{i=1}^{n} \phi_i,$$

   where $\phi_i$ is a statement denoting a *single* possible configuration of motifs that induces a positive biological readout.

   Each $\phi_i$ has the following form:

   $$\phi_i := (b_{m_{i,1}, p_{i,1}} \geq t_{i,1}) \wedge \cdots \wedge (b_{m_{i,d}, p_{i,d}} \geq t_{i,d}).$$

   That is, $\phi_i$ defines whether or not a specific configuration of $d \geq 1$ binding motifs $m_{i,1}, \ldots, m_{i,d}$ exist with sufficient strength at positions $p_{i,1}, \ldots, p_{i,d}$. Note that with finite sequences, all possible statements about combinations of motifs (expressable in first-order logic) can be written in this form, by the Disjunctive Normal Form Theorem.

Below, we proceed with our proof by constructing an instantiation of the ARGMINN architecture where the particular instantiation implements/recognizes a sentence $\sigma$.

**Motif-scanner module**

Given the above biophysical assumptions, the motif-scanner module's convolutional filters are sufficiently expressive to capture the binding strengths/likelihoods of each position for each potential binding motif.

We simply design this module to have $n_f$ filters equal to the number of unique motifs $m$ in $\sigma$, with width $w$ equal to the maximum width of any motif. We set the multiplicative weights of each filter with the PWM of each motif, so that this module precisely implements the PWM-scanning procedure

that is common in the field of regulatory genomics (Benos et al., 2002). For this proof, we set the bias to be 0. Note that if the convolutional filters $W$ are all PWMs, then the multiplicative weights are all non-negative, and so the pre-ReLU activation will also be non-negative.

Thus, the motif-scanner module outputs a set of binding strengths for each position of the sequence, for each possible motif. In other words, this module outputs $b_{m,p}$ for each relevant motif $m$, at each position $p$. This constitutes the motif activations $A$ of the model.

**Syntax-builder module**

Now we design an instantiation of the syntax-builder module which captures the logic in $\sigma$. In particular, our model will produce a pre-sigmoid output which is positive if and only if $\sigma$ is true.

The input to the syntax-builder module is the concatenation of motif activations $A$ (which is the matrix of binding strengths $b_{m,p}$) and the positional encodings $P$: $A\|P$. For ease of proof, let the positional encodings $P$ be a one-hot-encoded vector of size $\ell$, denoting position. We will use the notation $[A\|P]_p$ to denote the column vector at position $p$, which is an $(n_f + \ell)$-dimensional vector where the first $n_f$ entries contain the binding strengths/activations of each motif at position $p$, and the latter $\ell$ entries contain a one-hot encoding of position $p$.

Recall that within $\sigma$, $n$ is the number of statements in disjunction. Let $d_{max}$ be the maximum length of any of the $\phi_i$s. For simplicity of our proof and construction, we pad every $\phi_i$ to have exactly $d_{max}$ clauses by adding "dummy motif clauses". For example, let us pad with the clause $(b_{0,0} \geq 0)$. This simply checks that some arbitrary motif (index 0) at some arbitrary position (position 0) has non-negative activation, which will always be true since the convolutional weights in the motif-scanner module are PWMs.

We let the memory vector $m_l$ be of dimension $n(n_f + \ell)$. Additionally, we define the dimension of the query/key/value vectors be $n_f + \ell$.

Our proof (like many other similar proofs) relies on the universal approximation theorem of feed-forward networks (FFNs). We also leverage Lemma 19 from (Chiang et al., 2023), which shows that in an FFN, residual connections can effectively be ignored (which simplifies the construction of our network).

We proceed with induction on $d_{max}$, where our syntax-builder module has $d_{max}$ attention layers, with $n$ heads for each layer (one head for every $\phi_i$).

Base case: $d_{max} = 1$

Consider $d_{max} = 1$. In this case, each $\phi_i$ can be written as $\phi_i := b_{m_i,p_i} \geq t_i$. That is, $\phi_i$ is true if and only if motif $m_i$ at position $p_i$ is strong enough.

Now we instantiate a single-attention-layer syntax-builder module with $n$ attention heads. We repeat the following for each head independently:

For each head $i$, we learn to recognize $\phi_i$. Recall that the memory vector $m_0$ is initialized to be all 1s.

We define the weight matrices $W_{Q,1}, W_{K,1}, W_{V,1}$ separately for each attention head in this proof, knowing that the final weight matrices for the attention layer as a whole are obtained by a simple concatenatation operation over the heads. Let $W_{Q,1}$ map $m_0$ to a single vector where the first $n_f$ entries is a one-hot encoding where the 1 is in the position of motif $m_i$, and the latter $\ell$ entries is filled with a very negative constant $-C$ ($C >> 0$), except for the position $p_i$ (indexed from within these latter $\ell$ entries), which has a 1. Let $W_{K,1} = W_{V,1} = I_{n_f+\ell}$, the identity matrix of appropriate size:

$$q_1 = W_{Q,1}m_0 = [\ 0\ \ \cdots\ \ 0\ \ 1\ \ 0\ \ \cdots\ \ 0\ \ |\ \ -C\ \ \cdots\ \ -C\ \ 1\ \ -C\ \ \cdots\ \ -C\ ]^{\mathsf{T}}$$

where the 1 is at index $m_i$ in the left block, and the 1 is at index $p_i$ within the right block.

$$K_1 = W_{K,1}[A\|P] = [A\|P] \quad V_1 = W_{V,1}[A\|P] = [A\|P]$$

Next, we take the dot product of every key vector with the query vector. For the key vector $[A\|P]_{p_i}$ (originating from position $p_i$), the dot product will be $b_{m_i,p_i} + 1$ (the first $n_f$ entries contribute the

binding strength $b_{m_i,p_i}$, and the latter $\ell$ entries contribute the 1). For any other position $p \neq p_i$, the dot product will be $b_{m_i,p} - C$.

We select $C$ to be a large-enough magnitude such that after the softmax, the attention score at position $p_i$ will be approximately 1, and all other positions will effectively be 0. Thus, the final vector being passed to the feed-forward network (FFN) is equivalent to the value vector corresponding to position $p_i$ (which is equivalent to the motif activations/positional encoding at position $p_i$):

$$a_1 V_1 = [A\|P]_{p_i}$$

With all attention heads together, we obtain a concatenated vector of size $n(n_f + \ell)$, where every contiguous $i$th block of $n_f + \ell$ entries corresponds to $\phi_i$, and contains the vector $[A\|P]_{p_i}$.

We then design our FFN to map from this vector to our final memory stream $m_1$. In particular, this FFN will produce an output vector of the same size, where every contiguous $i$th block of $n_f + \ell$ entries contains all $-\frac{1}{n_f+\ell}$ if $b_{m_i,p_i} < t_i$, and all $\frac{n}{n_f+\ell}$ otherwise. We invoke Lemma 19 from Chiang et al. (2023) and the Universal Approximation Theorem to perform this step.

Our final linear projection layer (which takes $m_1$ and maps to an output prediction $\hat{y}$) has weights of all 1 and bias of 0.

Together, this ensures that the output $\hat{y} > 0$ if and only if there exists an $i$ such that $b_{m_i,p_i} > t_i$ (i.e., $\phi_i$ is true).

Base case: $d_{max} = 2$

We proceed with a similar structure as with the above base case. In the first layer, we define $W_{Q,1}, W_{K,1}, W_{V,1}$ identically as above. However, after the first attention layer, we design the FFN differently so that $m_1$ contains information about the *next* motif within $\phi_i$ to search for.

In particular, each $\phi_i$ is of the form $\phi_i := (b_{m_{i,1},p_{i,1}} \geq t_{i,1}) \wedge (b_{m_{i,2},p_{i,2}} \geq t_{i,2})$. In the first attention layer, we define the weight matrices $W_{Q,1}, W_{K,1}, W_{V,1}$ as above, so that the vector passed to the FFN is a concatenation of $[A\|P]_{p_{i,1}}$ for all $i$.

Here, we design the FFN so that it will produce $m_1$, where every contiguous $i$th block of $n_f + \ell$ entries contains all 1s if $b_{m_{i,1},p_{i,1}} \geq t_{i,1}$, and all 0s otherwise.

Next, the second attention layer will identify the second motif in each $\phi_i$. Again, we consider each attention head $i$ separately.

In this second layer, $W_{Q,2}$ produces a query vector for each head which is similar to that in the previous base case: a vector of $n_f + \ell$ entries where the first $n_f$ is all 0 except for the position of $m_{i,2}$, and the latter $\ell$ entries are all $-C$ except for the position $p_{i,2}$ (however, note that if the first motif $m_{i,1}$ was not found at position $p_{i,1}$, then the $i$th block of $m_1$ will be all 0s, and so the query vector will be also all 0s). Again, we let $W_{K,2} = W_{V,2} = I_{n_f+\ell}$:

$$q_2 = W_{Q,2} m_1 = [\ 0 \quad \cdots \quad 0 \quad 1 \quad 0 \quad \cdots \quad 0 \quad | \quad -C \quad \cdots \quad -C \quad 1 \quad -C \quad \cdots \quad -C \ ]^{\mathsf{T}}$$

where the 1 is at index $m_{i,2}$ in the left block, and the 1 is at index $p_{i,2}$ within the right block. If the first motif $m_{i,1}$ was not found, then this vector will be all 0s.

$$K_2 = W_{K,2}[A\|P] = [A\|P] \quad V_2 = W_{V,2}[A\|P] = [A\|P]$$

We then follow the same construction as with the previous base case, where the second FFN produces the final memory stream $m_2$ based on comparing each $b_{m_{i,2},p_{i,2}}$ to $t_{i,2}$: every contiguous $i$th block of $n_f + \ell$ entries contains all $-\frac{1}{n_f+\ell}$ if $b_{m_{i,2},p_{i,2}} < t_{i,2}$, and all $\frac{n}{n_f+\ell}$ otherwise. If, however, the first motif $m_{i,1}$ was not found, then the FFN will always output $-\frac{1}{n_f+\ell}$ for that block (in this case, the query vector is all 0s, and the latter $\ell$ entries of the $i$th block in the input to the FFN will be a smeared fraction rather than a one-hot encoding). The final projection layer will be the same as with the first base case, leading to the desired outcome.

Inductive case

We complete our inductive proof for a general $d_{max}$, as the construction of the architecture with $d_{max}$ layers is a straightforward extension from the base cases.

We assume that in the first $d_{max} - 1$ layers, the memory stream $m_{d_{max}-1}$ is structured as contiguous blocks of $n_f + \ell$ entries, where the $i$th block is such that it contains all 1s if and only if $b_{m_{i,j},p_{i,j}} \geq t_{i,j}$ for all $j < d_{max}$ and all 0s otherwise. We then structure our final $d_{max}$th layer of the attention mechanism similarly to the second layer in the base case of $d_{max} = 2$.

## A.2 PROOF OF COROLLARY 1.1

Here, we prove that there exists a configuration of motifs specified by first-order logic, which is not recognizable by ExplaiNN.

Consider a dataset of sequences defined by the presence of distinct motifs $A, B, C, D$. Every sequence has exactly two instances of such motifs. A positive sequence is defined by having both $A$ and $B$, or both $C$ and $D$. A negative sequence is defined by any other combination: $AC$, $BD$, $AD$, or $BC$. This constructed example is a realistic one, as it is the binding rule seen in transcription factors with two half sites (e.g., REST (Tang et al., 2021)), or co-factor binding where the motifs and transcription factors are unidirectional (e.g., JUND and TEAD).

Within ExplaiNN, each CNN "unit" learns the presence of one such motif, and returns a scalar score. Let there be four CNN units, one for each motif. For each input sequence, we obtain four such scores: $s_A, s_B, s_C, s_D$.

Each CNN unit consists of a single convolutional filter whose activation is maximized by the motif it learns. For simplicity, we assume that given a convolutional filter that learns substring $X$, the distribution of activations of that filter on background sequences is identical to the distribution of activations on non-$X$ motifs. This could be realized, for example, by motifs whose composition is base/letter-wise distributed identically to the background (e.g., uniform). Thus, for a convolutional filter which recognizes $X$, the distribution of its activations is identical across all non-$X$ substrings. Given sufficiently long sequences, the CNN unit for some motif $X$ will output a scalar score as follows: if $X$ is in the sequence, $s_X = p_X$, a score for "positives"; if $X$ is not present, $s_X = n_X \neq p_X$, a score for "negatives".

Given these CNN units, suppose it is possible to distinguish positive and negative sequences with a linear combination, as in ExplaiNN. The output of the model is $w_A s_A + w_B s_B + w_C s_C + w_D s_D + \beta$ for scalar weights $w_A, \ldots, w_D$ and bias $\beta$.

For the classifier to be sufficiently expressive, we require that positive examples have a final output that is at least some threshold $\tau$, and negative examples to have an output that is strictly less than $\tau$.

Thus, we have the following inequalities (one for each possible pair of motifs):

$$w_A p_A + w_B p_B + w_C n_C + w_D n_D + \beta \geq \tau$$
$$w_A n_A + w_B n_B + w_C p_C + w_D p_D + \beta \geq \tau$$
$$w_A p_A + w_B n_B + w_C p_C + w_D n_D + \beta < \tau$$
$$w_A p_A + w_B n_B + w_C n_C + w_D p_D + \beta < \tau$$
$$w_A n_A + w_B p_B + w_C p_C + w_D n_D + \beta < \tau$$
$$w_A n_A + w_B p_B + w_C n_C + w_D p_D + \beta < \tau$$

Adding inequalities of the same type:

$$w_A(p_A + n_A) + \cdots + w_D(p_D + n_D) \geq 2(\tau - \beta)$$
$$2w_A(p_A + n_A) + \cdots + 2w_D(p_D + n_D) < 4(\tau - \beta)$$

This is a contradiction, as this requires $w_A(p_A + n_A) + \cdots + w_D(p_D + n_D)$ to be both at least $2(\tau - \beta)$ and strictly less than $2(\tau - \beta)$.

Thus, ExplaiNN is not sufficiently expressive to capture every configuration of motifs expressable in first-order logic (e.g., exclusive disjunctions).

## B  Supplementary Figures and Tables

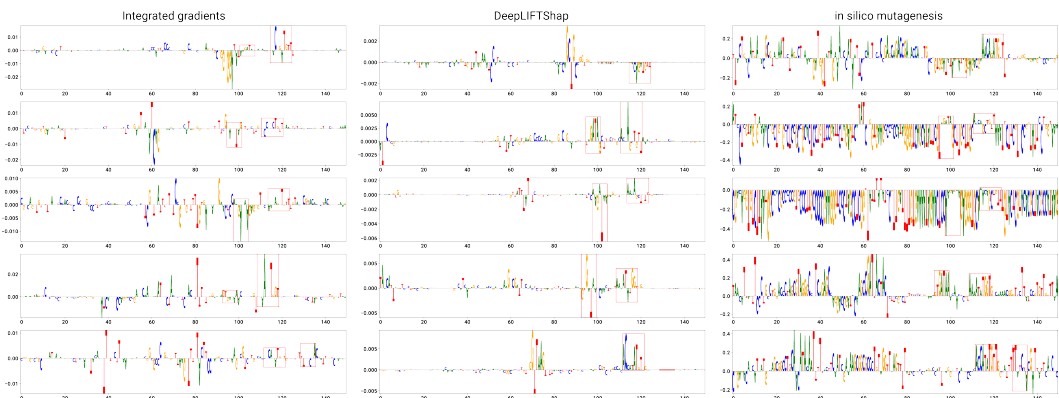

Figure S1: From a single standard CNN model trained on TAL/GATA binding, we show the attribution scores for the same five input sequences, computed using three different methods: integrated gradients (left), DeepLIFTShap (middle), and *in silico* mutagenesis (right). The locations of the true motifs are highlighted by the red box in each example. Although this model achieves near-perfect test accuracy, the importance scores remain unreliable and noisy. Regardless of the method, it is difficult to even identify *where* the motif is solely based on these score tracks, let alone *what* the motif is. Additionally, the methods disagree heavily with each other, even showing different signs (positive vs. negative) in importance for the true motifs.

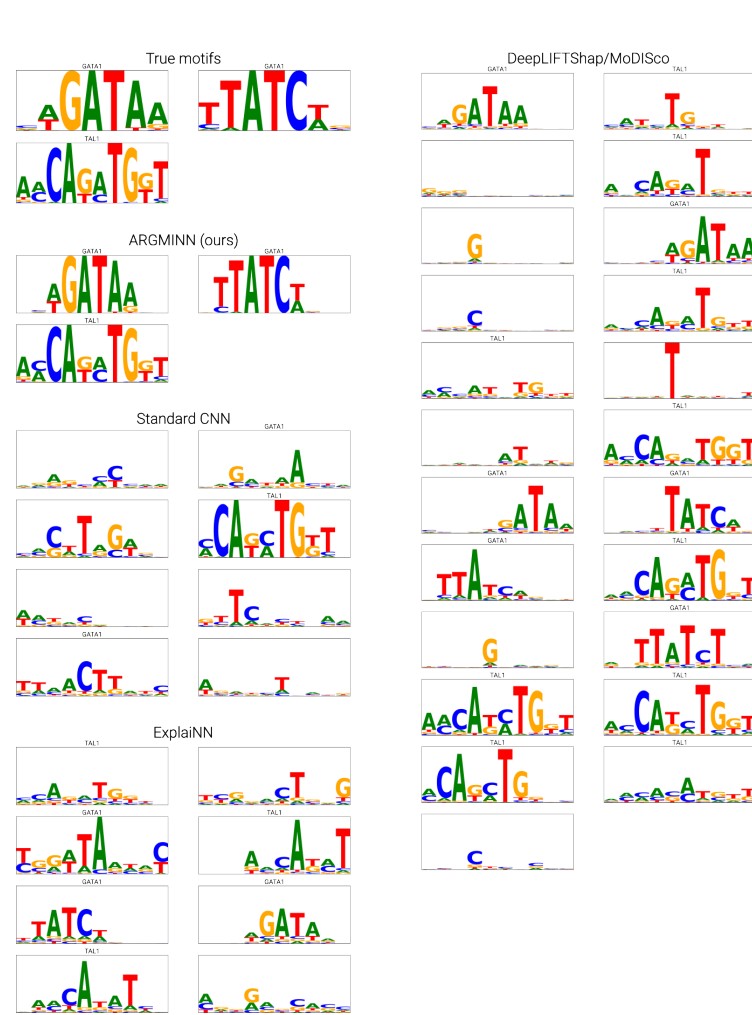

Figure S2: On a simulated TAL/GATA dataset, we show all motifs discovered by: 1) ARGMINN, 2) interpreting the first-layer filters of a standard CNN, 3) ExplaiNN, and 4) running MoDISco on DeepLIFTShap importance scores. Each motif is labeled with the most similar motif from the simulation, using TOMTOM. Motifs which are not sufficiently similar to any of the motifs in the simulation (as determined by TOMTOM's default thresholds), remain unlabeled. We also show the true motifs used in the simulation.

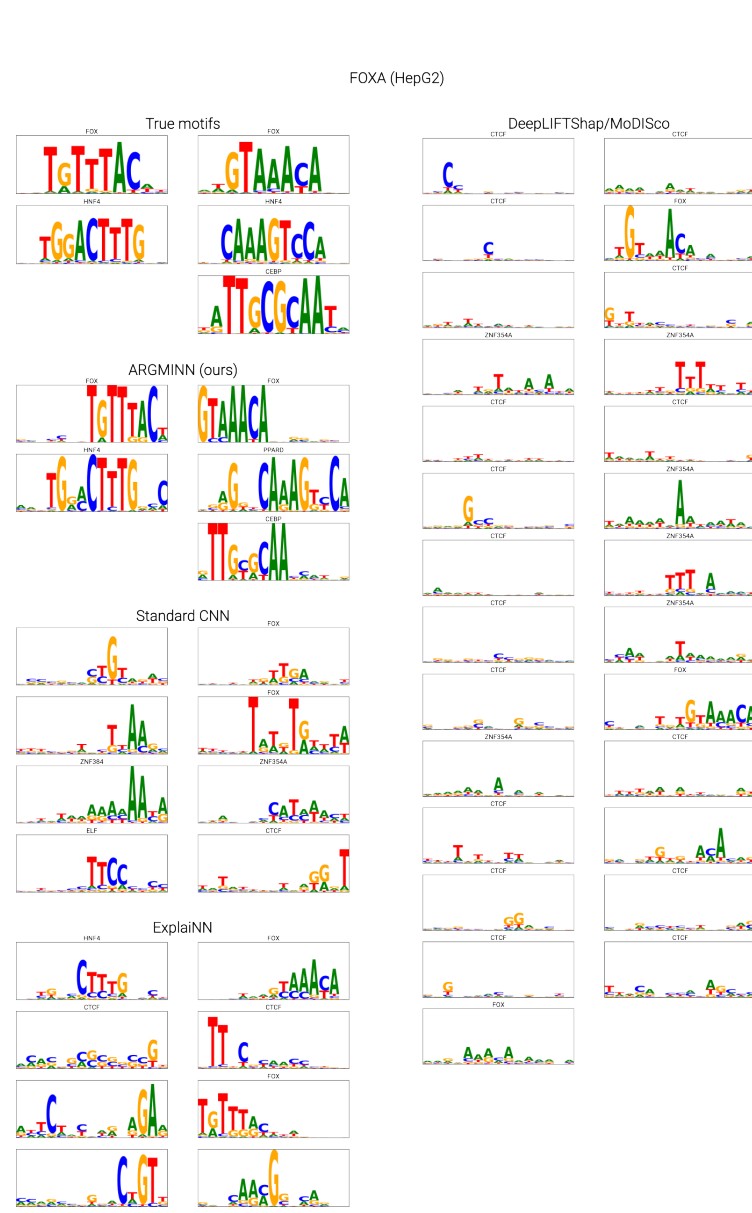

Figure S3: On an experimental dataset of FOXA2 binding in HepG2, we show all motifs discovered by: 1) ARGMINN, 2) interpreting the first-layer filters of a standard CNN, 3) ExplaiNN, and 4) running MoDISco on DeepLIFTShap importance scores. Each motif is labeled with the most similar known human motif, using TOMTOM. Motifs which are not sufficiently similar to any known human motif (as determined by TOMTOM's default thresholds), remain unlabeled. We also show ground-truth motifs from JASPAR which are supported by external literature.

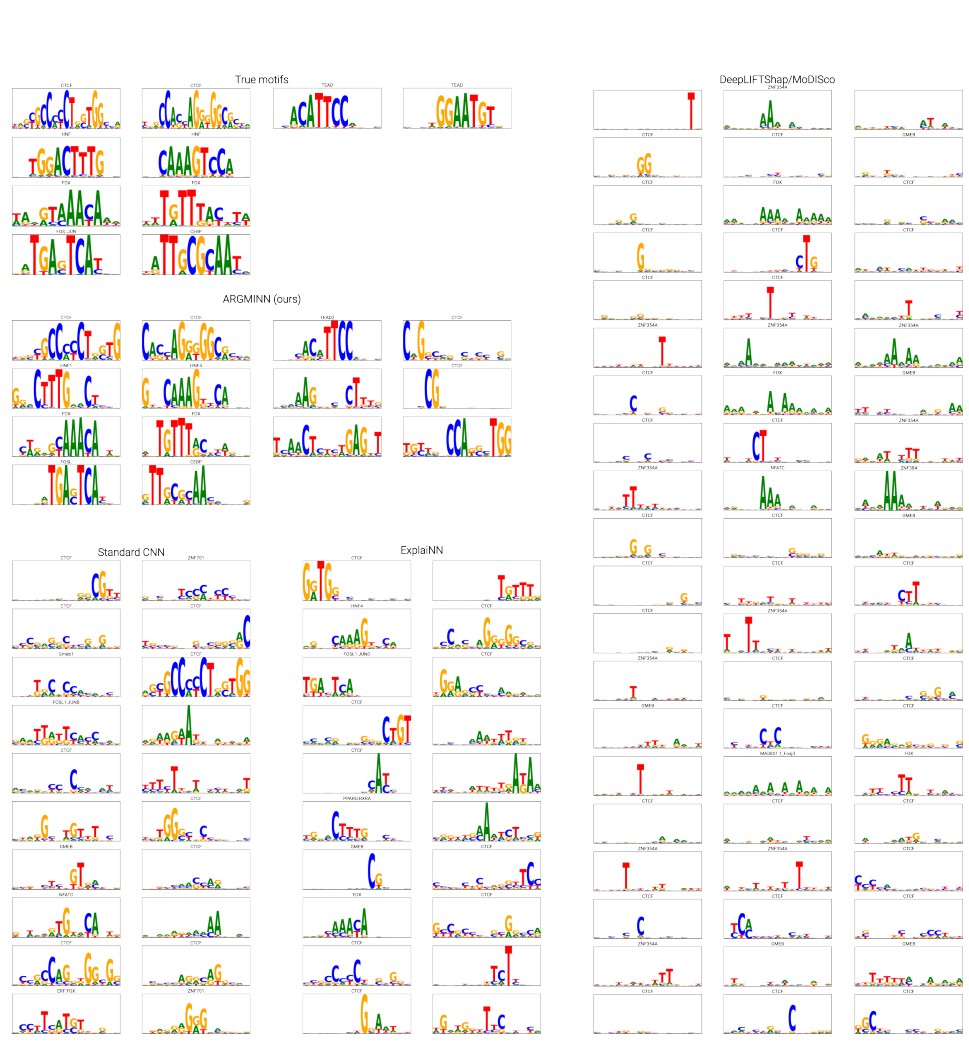

Figure S4: On an experimental dataset of DNase accessibility in HepG2, we show all motifs discovered by: 1) ARGMINN, 2) interpreting the first-layer filters of a standard CNN, 3) ExplaiNN, and 4) running MoDISco on DeepLIFTShap importance scores. Each motif is labeled with the most similar known human motif, using TOMTOM. Motifs which are not sufficiently similar to any known human motif (as determined by TOMTOM's default thresholds), remain unlabeled. We also show ground-truth motifs from JASPAR which are supported by external literature.

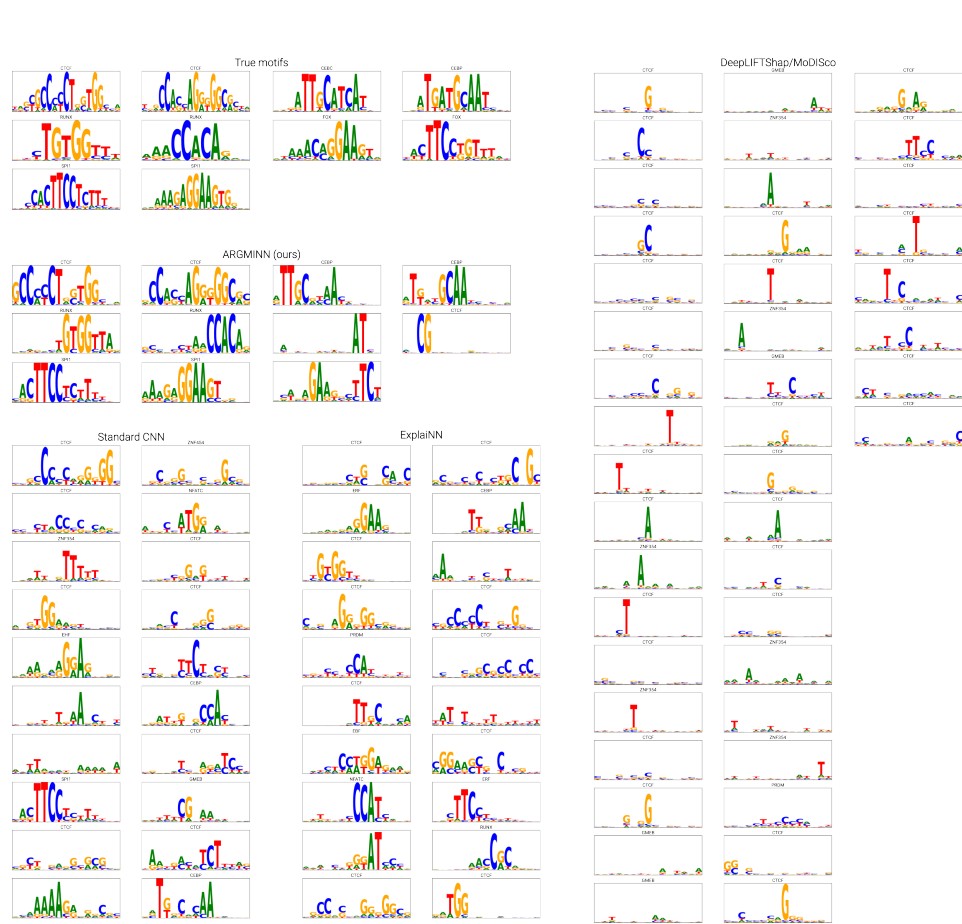

Figure S5: On an experimental dataset of DNase accessibility in HL-60, we show all motifs discovered by: 1) ARGMINN, 2) interpreting the first-layer filters of a standard CNN, 3) ExplaiNN, and 4) running MoDISco on DeepLIFTShap importance scores. Each motif is labeled with the most similar known human motif, using TOMTOM. Motifs which are not sufficiently similar to any known human motif (as determined by TOMTOM's default thresholds), remain unlabeled. We also show ground-truth motifs from JASPAR which are supported by external literature.

Table S1: Identification and redundancy of discovered motifs

| Dataset | Relevant motif | ARGMINN | Traditional CNN | ExplaiNN | DeepLIFTShap/MoDISco |
|---|---|---|---|---|---|
| SPI1 | SPI1 | **1** | 4 | 2 | 17 |
| TAL/GATA | TAL | **1** | **1** | 3 | 10 |
|  | GATA1 | **1** | **1** | 2 | 3 |
| E2F6 | E2F6 | **1** | 2 | 3 | 10 |
|  | MAX | **1** | **1** | 2 | 5 |
| JUND/TEAD | JUND-TRE | **1** | 3 | 3 | 22 |
|  | JUND-CRE | **1** | **1** | 2 | 5 |
| REST | REST-left | **1** | **1** | **1** | 9 |
|  | REST-right | **1** | **1** | 3 | 4 |
| SPI1/CTCF | SPI1 | **1** | 2 | 2 | 11 |
|  | CTCF | **1** | **1** | 2 | 2 |
| CTCF (HepG2) | CTCF | 2 | 4 | 3 | 12 |
|  | FOX | **1** | 3 | **1** | 7 |
| FOXA1 (HepG2) | HNF4 | **1** | 0 | **1** | 0 |
|  | CEBP | **1** | 0 | **1** | 0 |
|  | HNF | 2 | 0 | 2 | 0 |
|  | TEAD | **1** | 0 | 0 | 0 |
| DNase (HepG2) | CTCF | 3 | 7 | 6 | 25 |
|  | FOX | **1** | 3 | **1** | 7 |
|  | CEBP | **1** | 0 | 0 | 0 |
|  | FOS::JUN | **1** | 0 | **1** | 0 |
|  | RUNX | **1** | 0 | **1** | 0 |
| DNase (HL-60) | CTCF | 2 | 6 | 9 | 17 |
|  | FOX | 0 | 2 | **1** | 0 |
|  | CEBP | 2 | **1** | 2 | 0 |
|  | SPI1 | **1** | **1** | 0 | 0 |
|  | FOSL2::JUN | **1** | 2 | **1** | 0 |
| DNase (K562) | CTCF | 3 | 8 | 7 | 8 |
|  | GATA | **1** | **1** | 2 | **1** |

Number of times each relevant motif was discovered by each method. A value of 0 means the motif was not discovered at all. Values greater than 1 indicate redundancy.

Table S2: Similarity of discovered motifs to ground truth

| Dataset | Relevant motif | ARGMINN | Traditional CNN | ExplaiNN | DeepLIFTShap/MoDISco |
|---|---|---|---|---|---|
| SPI1 | SPI1 | **10.890** | 1.583 | 4.876 | 8.452 |
| TAL/GATA | TAL | 8.263 | 7.017 | 2.769 | **8.395** |
| | GATA1 | **7.914** | 1.813 | 4.140 | 5.737 |
| E2F6 | E2F6 | **8.031** | 6.547 | 5.335 | 6.699 |
| | MAX | **4.289** | 0.713 | 1.969 | 3.706 |
| JUND/TEAD | JUND-TRE | 7.164 | 2.616 | 4.584 | **7.298** |
| | JUND-CRE | 6.150 | 1.544 | 2.667 | **7.074** |
| REST | REST-left | **13.727** | 0.428 | 1.783 | 8.027 |
| | REST-right | **12.895** | 0.742 | 3.991 | 10.739 |
| SPI1/CTCF | SPI1 | **10.339** | 6.894 | 6.938 | 8.578 |
| | CTCF | **13.129** | 2.140 | 7.360 | 4.210 |
| CTCF (HepG2) | CTCF | **21.974** | 6.800 | 15.945 | 13.338 |
| | FOX | **5.651** | 2.712 | 4.917 | 5.017 |
| FOXA1 (HepG2) | HNF4 | **7.188** | 0 | 4.587 | 0 |
| | CEBP | **5.688** | 0 | 1.083 | 0 |
| | HNF | **6.178** | 0 | 5.852 | 0 |
| | TEAD | **8.545** | 0 | 0 | 0 |
| DNase (HepG2) | CTCF | **25.482** | 24.543 | 19.120 | 4.262 |
| | FOX | **6.831** | 1.270 | 3.815 | 2.028 |
| | CEBP | **4.797** | 0 | 0 | 0 |
| | FOS::JUN | **9.686** | 0 | 1.982 | 0 |
| | RUNX | **4.660** | 0 | 3.232 | 0 |
| DNase (HL-60) | CTCF | **24.428** | 6.377 | 16.945 | 5.423 |
| | FOX | 0 | 1.149 | **4.178** | 0 |
| | CEBP | **4.947** | 4.736 | 4.168 | 0 |
| | SPI1 | **11.434** | 7.568 | 0 | 0 |
| | FOSL2::JUN | **7.646** | 7.338 | 5.198 | 0 |
| DNase (K562) | CTCF | **23.810** | 6.121 | 13.568 | 1.776 |
| | GATA | **9.159** | 0.533 | 7.078 | 0.345 |

Similarity of closest motif discovered by each method to each relevant motif. If a method did not discover a motif, it is given a similarity of 0.

Table S3: Extraneous discovered motifs

| Dataset | ARGMINN | Traditional CNN | ExplaiNN | DeepLIFTShap/MoDISco |
|---|---|---|---|---|
| SPI1 | **0** | 1 | 4 | 10 |
| TAL/GATA | **0** | 5 | 2 | 7 |
| E2F6 | **0** | 3 | 2 | 4 |
| JUND/TEAD | **0** | 4 | 3 | 2 |
| REST | **0** | 6 | 1 | 7 |
| SPI1/CTCF | **0** | 3 | 1 | 13 |
| CTCF (HepG2) | **0** | 1 | 3 | 17 |
| FOXA1 (HepG2) | 2 | 3 | 3 | 18 |
| DNase (HepG2) | 3 | **2** | 4 | 12 |
| DNase (HL-60) | 2 | 3 | **2** | 11 |
| DNase (K562) | 2 | 3 | **0** | 41 |

Number of extraneous motifs (i.e., those which do not match any known relevant motif) found by each method for each dataset.

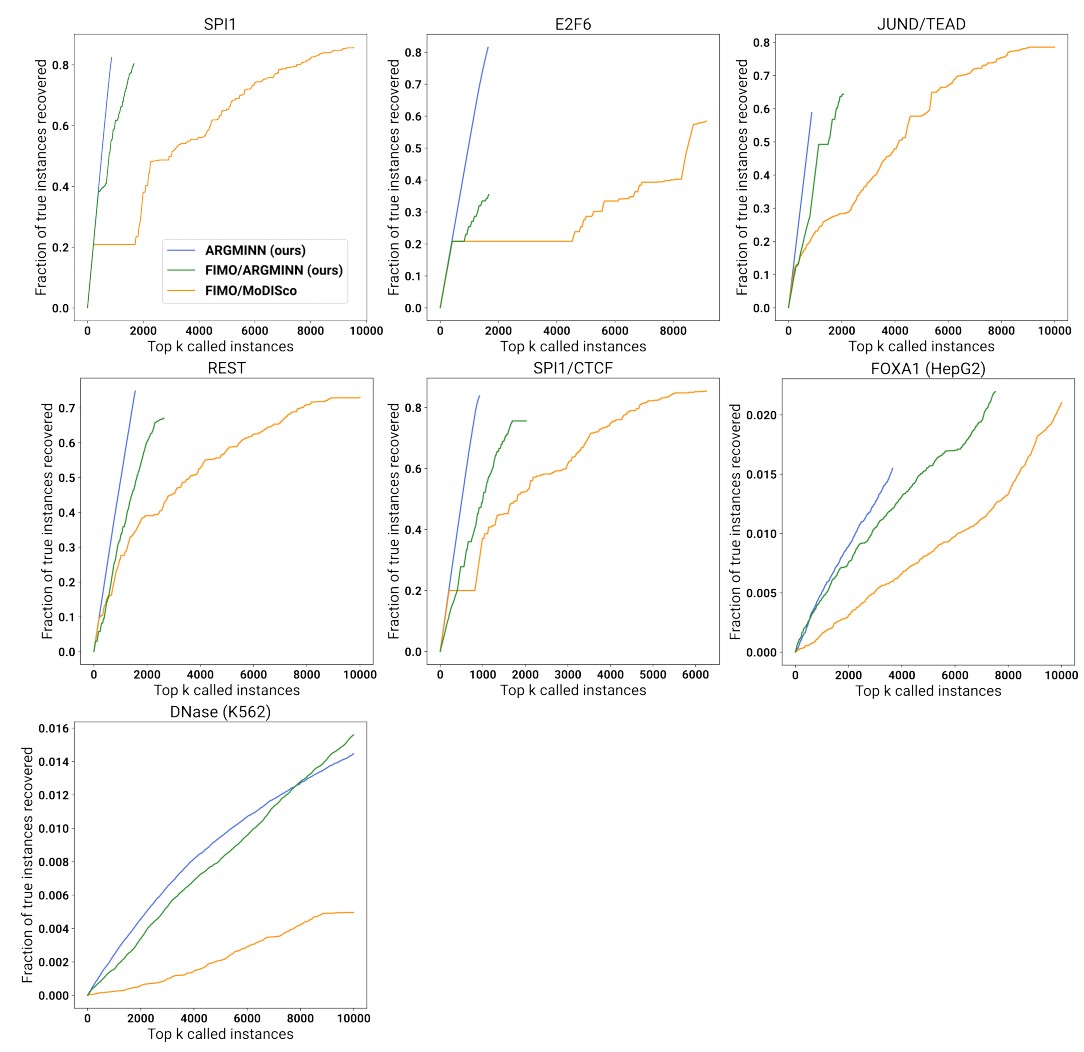

Figure S6: We show additional examples of motif-instance quality, comparing ARGMINN with the traditional method of sequence-scanning using FIMO. As before, We rank motif instances by confidence (attention score from ARGMINN, or FIMO hit q-value), and compute the fraction of true motif instances that are covered in a top-$k$ fashion. We also compare our method to using FIMO to scan with ARGMINN-discovered motifs.

Table S4: Motif-instance precision and recall

| Dataset | **Precision** | | | **Recall** | | |
|---|---|---|---|---|---|---|
| | A (ours) | F/AM (ours) | F/MM | A (ours) | F/AM (ours) | F/MM |
| SPI1 | **0.953** | 0.881 | 0.852 | 0.824 | 0.804 | **0.858** |
| TAL/GATA | **0.939** | 0.896 | 0.857 | **0.703** | 0.410 | 0.532 |
| E2F6 | **0.954** | 0.804 | 0.777 | **0.816** | 0.355 | 0.584 |
| JUND/TEAD | **1.000** | 0.920 | 0.980 | 0.588 | 0.644 | **0.785** |
| REST | **0.972** | 0.853 | 0.891 | **0.749** | 0.671 | 0.731 |
| SPI1/CTCF | **0.909** | 0.739 | 0.807 | 0.838 | 0.756 | **0.854** |
| CTCF (HepG2) | 0.291 | 0.355 | **0.399** | **0.038** | 0.033 | 0.028 |
| FOXA1 (HepG2) | 0.190 | **0.201** | 0.153 | 0.016 | **0.022** | 0.021 |
| DNase (HepG2) | 0.229 | **0.386** | 0.196 | **0.014** | 0.012 | 0.001 |
| DNase (K562) | 0.328 | **0.468** | 0.315 | 0.014 | **0.016** | 0.005 |

Precision and recall values for motif instances discovered by ARGMINN (A) and by FIMO. We initialize FIMO either with motifs discovered by ARGMINN (F/AM), or with motifs discovered by MoDISco (F/MM).

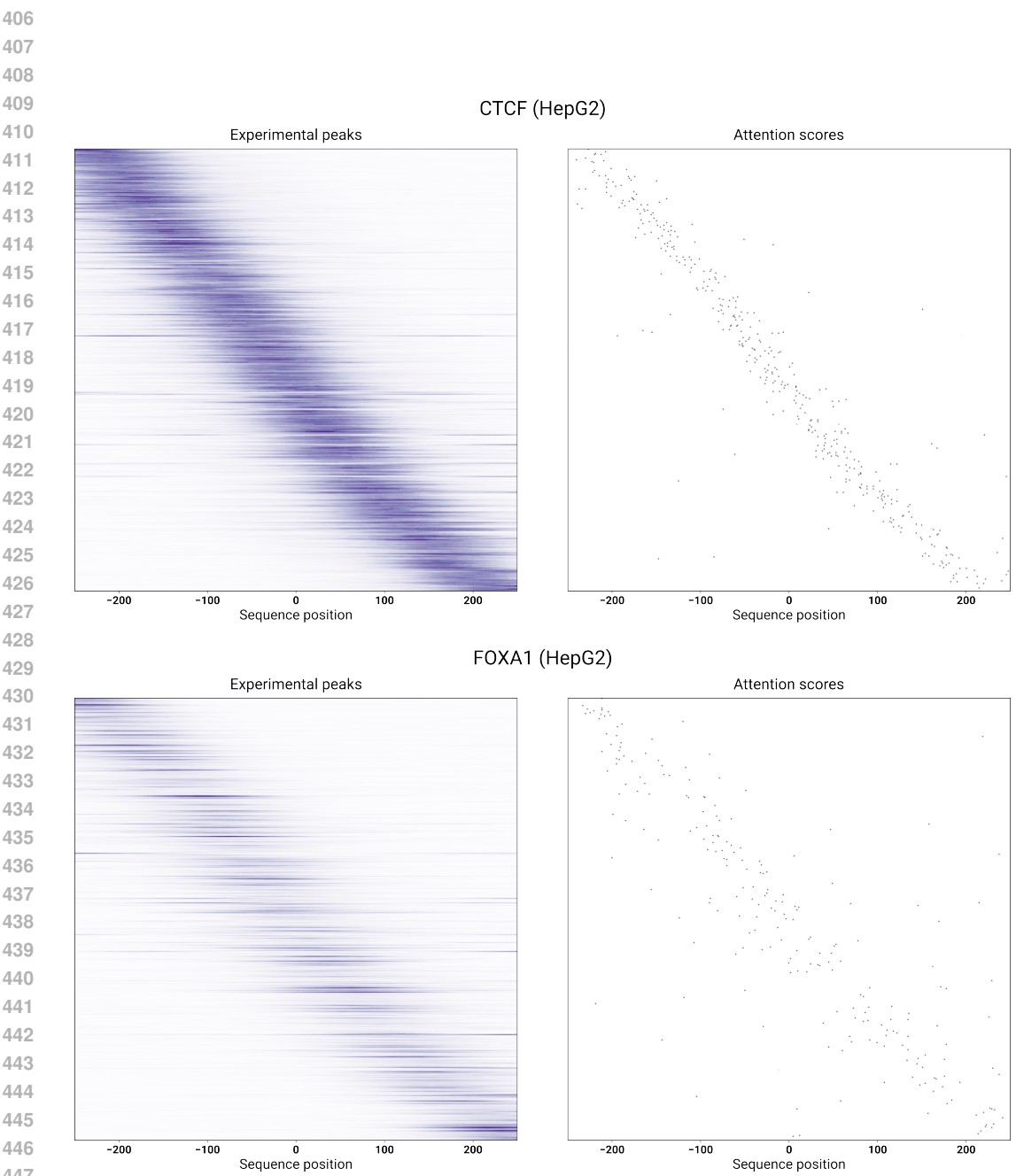

Figure S7: We show the attention scores and the strength of the experimentally determined peaks at the test-set sequences for CTCF in HepG2 (above) and FOXA1 in HepG2 (below). In order to avoid center bias, the test peaks were independently and randomly jittered by up to 200 bp in either direction with uniform probability. The peaks/sequences are ordered left to right by the jittered peak summit. The attention scores closely track the peak locations across sequences, which demonstrates biological support for the interpretability of the attention scores.

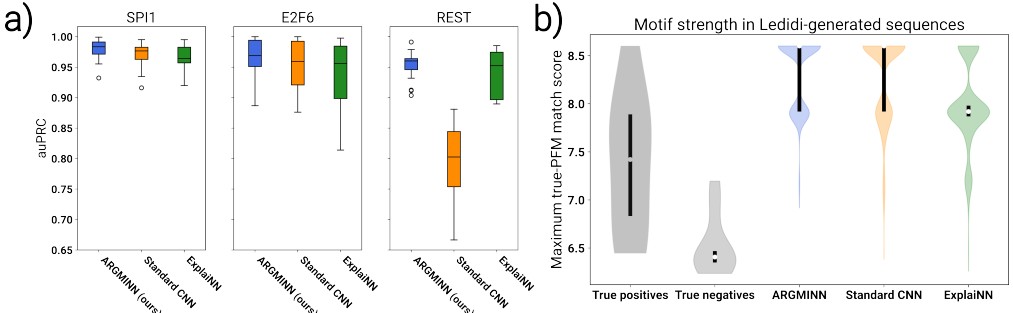

Figure S8: **a)** After training on 50% background GC content, we tested ARGMINN, a standard CNN, and ExplaiNN on varying levels of GC content, and show the resulting distribution of predictive performance. **b)** We used Ledidi to perform back-propagation-based sequence design on models trained to predict SPI1 binding, generating novel sequences which are meant to maximize the likelihood of SPI1 binding. We evaluated the quality of the generated sequences from each model by quantifying the distribution of match scores to the true SPI1 motif.

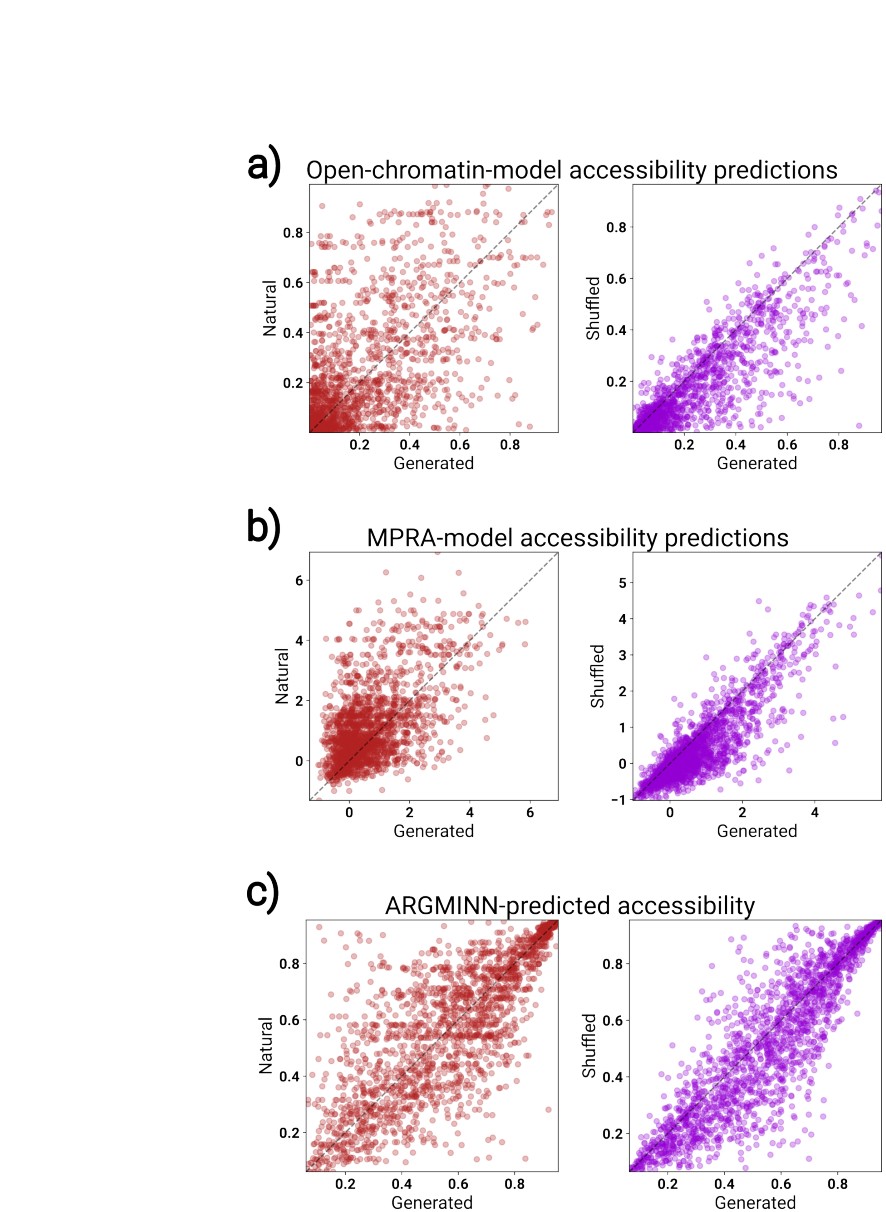

Figure S9: We performed additional *in silico* validation of our interpretably designed HepG2-accessible sequences. We tested our generated sequences using multiple independently trained models (trained on different datasets) as oracles: **a)** an open-chromatin model trained on an independently collected dataset of HepG2 accessibility; and **b)** an MPRA model trained on independently collected massively-parallel-reporter-assay data. We also show the predictions made by ARGMINN using the model as a predictive oracle rather than an interpretable-design agent (**c**)). In all cases, the generated sequences were far more accessible than background sequences, and were about as accessible as natural sequences identified by the experiment.

Table S5: Model performance

| Dataset | Accuracy | | | auROC | | | auPRC | | |
|---|---|---|---|---|---|---|---|---|---|
| | A (ours) | SC | E | A (ours) | SC | E | A (ours) | SC | E |
| SPI1 | **0.898** | 0.887 | 0.895 | **0.955** | 0.949 | 0.951 | **0.958** | 0.957 | 0.954 |
| TAL/GATA | **0.884** | 0.838 | 0.856 | **0.954** | 0.910 | 0.939 | **0.959** | 0.918 | 0.941 |
| E2F6 | **0.908** | 0.895 | 0.871 | **0.971** | 0.948 | 0.944 | **0.975** | 0.952 | 0.944 |
| JUND/TEAD | **0.988** | 0.981 | 0.984 | **0.999** | 0.997 | 0.999 | **0.999** | 0.997 | 0.999 |
| REST | **0.883** | 0.771 | 0.856 | **0.947** | 0.835 | 0.929 | **0.940** | 0.824 | 0.927 |
| SPI1/CTCF | **0.909** | 0.860 | 0.900 | **0.968** | 0.931 | 0.955 | **0.965** | 0.928 | 0.953 |
| CTCF (HepG2) | 0.769 | 0.775 | **0.782** | 0.848 | 0.850 | **0.859** | 0.854 | 0.852 | **0.860** |
| FOXA1 (HepG2) | 0.728 | **0.734** | 0.730 | 0.805 | 0.809 | **0.810** | 0.798 | **0.806** | 0.804 |
| DNase (HepG2) | 0.730 | 0.730 | **0.741** | 0.816 | 0.812 | **0.824** | 0.815 | 0.809 | **0.818** |
| DNase (HL-60) | 0.753 | 0.760 | **0.769** | 0.833 | 0.842 | **0.853** | 0.826 | 0.839 | **0.847** |
| DNase (K562) | 0.762 | 0.768 | **0.774** | 0.848 | 0.852 | **0.860** | 0.845 | 0.846 | **0.853** |

Model performance comparing ARGMINN (A), a standard CNN (SC), and ExplaiNN (E). All models were given the same number and size of convolutional filters, and the overall complexity/capacity of the models were kept as similar as possible for comparison.

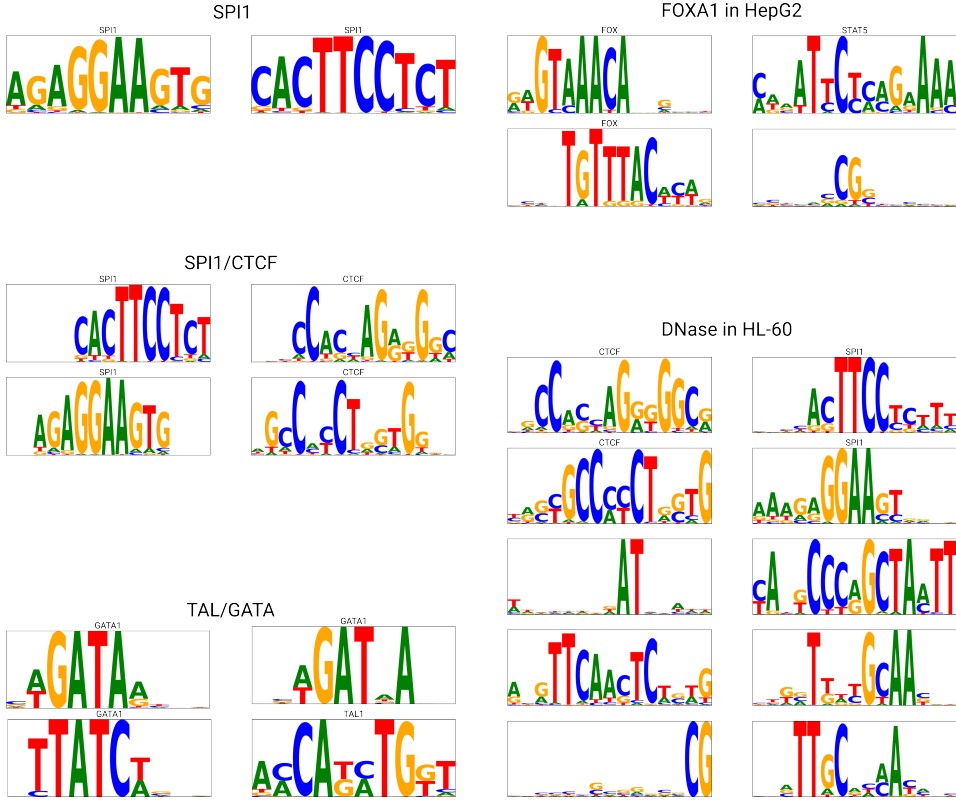

Figure S10: We applied our filter regularizer (Equation 2, Equation 4) to the standard CNN architecture, and show the resulting motifs extracted from the first-layer filters. With the filter regularization, the standard CNN attained the ability to show clean, non-redundant, relevant motifs in its filters. Thus, our filter regularizer is able to turn even traditional neural networks into more mechanistically interpretable architectures. Importantly, however, although a standard CNN with our filter regularizer can now reveal discovered motifs, without our unique attention mechanism (Equation 3), it still is unable to easily reveal motif instances and syntax. As such, these partially interpretable architectures would still need to rely on traditional motif-instance-scanning algorithms.

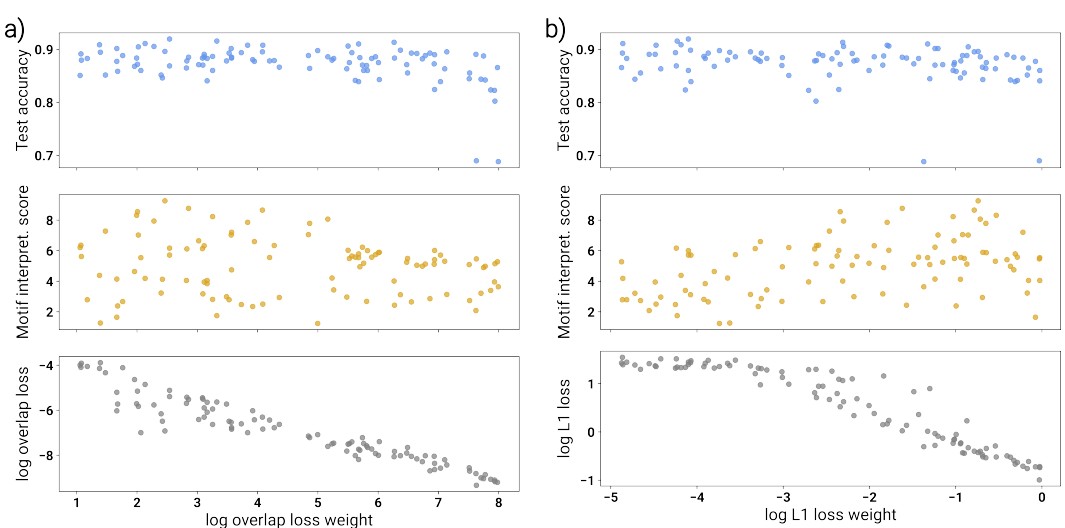

Figure S11: We show the effect of the filter-overlap (left) and filter-L1 loss (right) weights on the performance and interpretability of ARGMINN. We also show the effect of the loss weights on the value of the filter-overlap and filter-L1 loss itself. Interpretability is measured by the maximum similarity of the discovered motifs to the true motif. In general, ARGMINN's predictive performance and interpretability remained largely invariant to the loss weights over a wide range of orders of magnitude.

Table S6: Space/time requirements of different architectures

| Model | Training time (min) | Number of parameters |
|---|---|---|
| ARGMINN | 15 | 13000 |
| Standard CNN | 6 | 2100 |
| ExplaiNN | 8 | 58000 |

We show the training time and number of parameters in each of our models for a typical example (in this case, our SPI1 dataset). Note that the increased training time for ARGMINN stems chiefly from the computation of the filter-overlap regularizer. The increased parameter count for ExplaiNN is due to the separation of the architecture into individual CNN towers.

## C  SUPPLEMENTARY METHODS

All of the code used to generate the results and figures in this paper is available here:

`https://github.com/Genentech/ARGMINN`

### C.1  TRAINING DATA

For our simulated datasets, we downloaded motif PFMs (position frequency matrices) from JASPAR (Fornes et al., 2020), and trimmed off low-information-content flanks. When training and testing, we randomly generated 500 bp sequences on the fly. Motif instances were sampled from the PFMs, and inserted into a randomly sampled background (from a uniform distribution of A, C, G, and T). Motifs (or combinations of them) were randomly inserted in the central 100 bp of the background. Our simulations contained motif configurations as follows:

- SPI1: all positive sequences have a single instance of the SPI1 motif

- TAL/GATA: 37.5% of positive sequences have a single instance of TAL1, 37.5% have a single instance of GATA1, and 25% of sequences have both (either 7, 8, or 9 bp apart)

- E2F6: 10% of positive sequences have a single instance of E2F6, and 90% have both E2F6 and MAX, between 30 and 60 bp apart; 50% of negative sequences have only the MAX motif, and the other 50% are random background

- JUND: 25% of positive sequences have the JUND TRE motif, 25% have the JUND CRE motif, and 50% have the JUND TRE motif followed by the TEAD4 motif with a spacing of 6 bp in between

- REST: all positive sequences have both the left- and right-half motifs, with a spacing of 2, 6, 7, 8, 9, 10, or 11 bp apart; 25% of negative sequences have only the left-half motif, 25% have only the right-half motif, and 50% are random background

- SPI1/CTCF: all positive sequences have a single instance of the SPI1 motif; 50% of negative sequences have SPI1 followed by the CTCF motif (either 30, 40, or 50 bp apart), and 50% are random background

When generating simulated sequences, we scanned the random backgrounds for spurious matches and filtered out such instances. Note that this is a crucial step for simulated datasets, as for short motifs such as GATA (6 bp), we would expect over 12% of random backgrounds to contain a perfect match by chance. This is an issue which is much rarer in real datasets, but for our simulations, we scanned the PFMs (and their reverse complements) across our generated backgrounds (for positive and negative sequences), and ensured that no PFM which was used in the simulation attained a match score of over 0.9.

For our experimental datasets, we downloaded the IDR peaks from ENCODE (Consortium, 2012) for the following experiments:

Table S7: Experimental ENCODE datasets

| Dataset | ENCODE experiment ID | IDR peaks file ID |
|---|---|---|
| CTCF (HepG2) | ENCSR607XFI | ENCFF664UGR |
| FOXA2 (HepG2) | ENCSR865RXA | ENCFF081USG |
| REST (K562) | ENCSR054JMQ | ENCFF118ECK |
| DNase (HepG2) | ENCSR149XIL | ENCFF897NME |
| DNase (HL-60) | ENCSR889WKL | ENCFF773SFA |
| DNase (K562) | ENCSR000EKS | ENCFF274YGF |
| DNase (GM12878) | ENCSR000EMT | ENCFF073ORT |

Our positive dataset consisted of random 500 bp sequences drawn from the genome, where at least half of the 500 bp overlaps a peak (if the peak is less than 500 bp), or at least half of the peak overlaps the 500 bp (if the peak is over 500 bp). Our negative dataset consisted of randomly sampled intervals from the genome. If a randomly selected negative sample overlapped a peak by more than half (or *vice versa*, it was relabeled as positive for the batch).

Note that when training, we automatically used reverse-complement augmentation so that every batch contained sequences along with their reverse complements.

## C.2 MODEL ARCHITECTURES

Our ARGMINN architecture consists of two modules: motif scanners and a syntax builder. The motif scanners consist of a single convolutional layer of $n_f$ filters, each of $w$ bp in width (typical values are $n_f = 8$, $w = 10$), which scan across a one-hot-encoded DNA sequence. The result is passed to a ReLU, and the output constitutes the "motif activations".

The motif activations are concatenated with a positional encoding of dimension $d = 16$. Our positional encoding is defined as follows:

$$P_{i,2j} = \sin(\frac{i}{10000^{\frac{2j}{d}}}), P_{i,2j+1} = \cos(\frac{i}{10000^{\frac{2j}{d}}})$$

where $i$ is the position along the sequence, and $j \in \{0, \ldots, \frac{d}{2} - 1\}$.

Let $A\|P$ be the motif activations and positional encodings concatenated together. This is passed to two consecutive memory-stream-based attention layers. The $l$th attention layer starts with a memory stream $m_{l-1}$ of dimension 128 ($m_0$ is a vector of all 1s), and $A\|P$. In each layer, $m_{l-1}$ is passed through a linear layer to obtain a single query $q_l$ of dimension equal to the dimension of each input token in $A\|P$. Two separate linear layers also convert each token in $A\|P$ into a set of key vectors and value vectors (of the same dimension as the input token). The query, keys, and values are reshaped to obtain 4 attention heads. We then compute the attention scores by multiplying the query against all keys, and normalizing by $\sqrt{d_{q_l}}$, where $d_{q_l}$ is the dimension of the query/key/value vectors. We then perform dropout on the attention scores with dropout rate 0.1, and softmax the scores for each attention head. These attention scores are used to weight the value vectors in a weighted sum, which is then reshaped to reincorporate the heads. This is passed through another linear layer which retains the same dimension, followed by dropout and layer normalization. This is then fed to a 2-layer MLP with ReLU and dropout in between the two linear layers, mapping the result to the same dimension as $m_{l-1}$. After a final dropout and layer norm, this is added to $m_{l-1}$ to obtain $m_l$.

After all attention layers, the final $m_l$ is passed to a single linear layer which maps it to a scalar prediction which is passed to a sigmoid activation function (for binary prediction).

Our standard CNN follows an architecture which is common in the literature for single-task predictions. We apply 3 successive convolutional layers to the input sequence, each with $n_f$ filters. The filters of the first layer have width $w$. The next two convolutional layers have filters of width 5. After each layer we apply ReLU and batch normalization. We then perform max pooling with a filter size of 40 and a stride of 40. This is passed to two linear layers of 10 and 5 hidden dimensions each. After each linear layer, we apply ReLU and batch normalization. Finally, a final linear layer maps the result to a sigmoid-activated output.

Our ExplaiNN implementation follows the description in Novakovsky et al. (2022b). As with the other architectures, we use the same number and width of first-layer convolutional filters.

Depending on the complexity of the dataset's syntax, we selected the ARGMINN architectural hyperparameters accordingly. Note that these values were not tuned at all, and were chosen at the outset based on domain knowledge and never modified. We also ensured that for each dataset, we always used the same number and length of first-layer convolutional filters ($n_f$ and $w$, respectively) in other architectures for a fair comparison.

Table S8: ARGMINN architectural hyperparameters

| Dataset | Number of attention layers $n_L$ | Number of filters $n_f$ | Length of filters $w$ |
|---|---|---|---|
| SPI1 | 1 | 8 | 10 |
| TAL/GATA | 1 | 8 | 10 |
| E2F6 | 2 | 8 | 10 |
| JUND | 1 | 8 | 10 |
| REST | 2 | 8 | 10 |
| SPI1/CTCF | 2 | 8 | 15 |
| CTCF (HepG2) | 1 | 8 | 15 |
| FOXA2 (HepG2) | 2 | 8 | 15 |
| REST (K562) | 2 | 8 | 8 |
| DNase (HepG2) | 3 | 20 | 15 |
| DNase (HL-60) | 3 | 20 | 15 |
| DNase (K562) | 3 | 20 | 15 |
| DNase (GM12878) | 3 | 20 | 15 |

## C.3 TRAINING SCHEDULES

We trained all of our models and performed all analyses on a single Nvidia A100.

When training, we used a batch size of 128 with an equal number of positives and negatives (this includes the reverse-complement augmentation). For our simulated datasets, each training epoch consisted of 100 batches, and each validation and test epoch consisted of 10 batches. For our experimental datasets, we reserved chr8 and chr10 for validation, and chr1 for test (all other autosomes along with chrX were used for training).

We trained all of our models for 40 epochs (regardless of architecture), and noted that the loss had converged in all cases. We used a learning rate of 0.001.

For ARGMINN, we weighted the secondary losses as follows:

- $\lambda_o$: 0 for the first 10 epochs, increasing from $10^{0.5}$ to $10^4$ evenly in logarithmic space over the next 20 epochs, and stable at $10^4$ for the last 10 epochs
- $\lambda_l$: 0 for the first 10 epochs, increasing from $10^{-4}$ to $10^{-3}$ evenly in logarithmic space over the next 20 epochs, and stable at $10^{-3}$ for the last 10 epochs

For all of our models (ARGMINN, standard CNN, ExplaiNN), we trained 3 random initializations and selected the one with the best test accuracy for downstream analyses.

## C.4 ANALYSES

**Extracting motifs from convolutional filters**

To extract motif PFMs from convolutional filters, we adopted a procedure based on that described in Kelley et al. (2016). Specifically, we passed the test set through the model and computed the average of all sub-sequences (pooling together all possible sub-sequences in the test set) which activated that filter to at least 50% of the maximum activation achieved over all such windows.

**Computing importance scores**

We computed importance scores using DeepLIFTShap, integrated gradients, or *in silico* mutagenesis (Shrikumar et al., 2017; Sundararajan et al., 2017).

For DeepLIFTShap and integrated gradients, we used PyTorch Captum. For DeepLIFTShap, we used a reference of 10 baselines consisting of dinucleotide-shuffled sequences, as recommended in Shrikumar et al. (2017). We also recovered the hypothetical importance scores at each position, as recommended for MoDISco (Shrikumar et al., 2018). For integrated gradients, we used a baseline of all 0s.

To obtain *in silico* mutagenesis scores, we computed the importance of a base by first computing the difference between the output prediction of the original sequence versus every possible mutation that

made at that position. We mean normalized over the base dimension to obtain a set of hypothetical importance scores. The actual importance score for a position was simply the hypothetical importance score (after mean normalization) for the base actually present in the sequence at the position.

**Discovering motifs with MoDISco**

In general, we discovered motifs using MoDISco from the standard CNN.

We computed importance scores over the entire positive-labeled dataset using the DeepLIFTShap algorithm as described above. We then ran MoDISco-lite v2.2.0 (Shrikumar et al., 2018) using a maximum of 10000 seqlets and default parameters, as recommended by the authors.

**Evaluating discovered motifs**

To compute matches to known motifs, we used TOMTOM (Bailey et al., 2015) to compute the q-value similarity between a PFM to known motifs (across all possible alignments). We reported the $-\log_{10}(q)$ value as similarity. For simulated datasets, we reported the closest match (i.e., highest similarity) to any motif in the dataset. For experimental datasets, we reported the closest match to any relevant motif in the JASPAR human-motif database (Fornes et al., 2020).

Note that to ensure a fair comparison, the lengths of the motifs were kept the same in each dataset. For ARGMINN, the standard CNN, and ExplaiNN, all models are trained with the same first-layer filter sizes. MoDISco by default outputs longer patterns, so we trimmed MoDISco-discovered motifs to the same size as the filters used by other methods (maximizing the total information content in the post-trimmed window; we used a uniform background for computing information content).

For our analyses on motif accuracy/similarity, redundancy, and number of extraneous motifs, we needed to match each motif discovered by each method to the closest *relevant* motif (or none at all). To do this, we first needed to identify the set of possible relevant motifs for each dataset (i.e., the set of motifs which would be considered biologically "accurate" for the task). For simulated datasets, the set of relevant motifs was simply the PFMs used to create the simulation. For experimental datasets, the set of relevant motifs was defined by first running TOMTOM against all known human motifs and pooling together the top matches (by motif family) over all methods and architectures (e.g., ARGMINN, MoDISco, etc.). We used a q-value cut off of 0.5. After pooling together all the top TOMTOM matches over all methods, we extracted a set of relevant motifs or motif families by checking for supporting literature. Any motif/family with supporting literature was kept as a relevant motif.

Finally, once the set of relevant motifs for each dataset was determined, we matched each motif (discovered by each method) to the closest relevant motif using TOMTOM. Discovered motifs which did not match any relevant motif (using the default TOMTOM threshold) were considered extraneous.

To compute redundancy, we counted the number of times each relevant motif was matched to by a method's discovered motifs. We kept track of forward and reverse-complement orientations. As long as one orientation was discovered, we considered that motif to had been found; we computed redundancy as the maximum number of times a relevant motif was matched to (maximum over orientations). For reverse-complement symmetric motifs/families, we did not consider orientations separately, and computed redundancy accordingly.

To compute motif accuracy, we computed the similarity (measured by the TOMTOM q-value) of the closest motif discovered by each method to each relevant motif.

**Tracing back motif instances and syntax**

To trace back motif instances for a particular input sequence in ARGMINN, we performed a forward pass and retained the motif activations and attention scores. Over all layers and all attention heads, we examined the positions in the sequence which had an attention score of at least 0.9. We then called a motif hit if the activation for a filter at that position was at least the average activation (for that filter) over the test set.

In our analyses, we called motif instances over the test set.

To identify syntax, we separated the input sequences by which motif instances were called, and computed the distribution of the spacings between the motif instances.

In order to rank motif instances from ARGMINN, we ranked by maxmimum attention score over all heads/layers at that position. To break any ties, we used the highest motif-filter activation score at that position.

### Scanning for motif instances with FIMO

Before running FIMO on MoDISco or ARGMINN motifs, we trimmed and filtered the motifs for high-information-content regions. Specifically, we cut off flanks with information content lower than 0.2. We then required that after trimming, the motif was at least 5 bp and had an average information content of 1.0. We used a uniform background for computing information content.

To scan for motif instances using FIMO, we started with PFMs and ran FIMO v5.0.5 (Bailey et al., 2015) on test-set sequences and their reverse complements. We used the default FIMO parameters.

To rank FIMO hits, we used the q-value from FIMO. We also collapsed overlapping FIMO hits before analyzing, keeping the most significant q-value between overlapping hits for ranking purposes.

### Evaluating motif instances

To evaluate our motif instances, we compared called motif instances to ground-truth instances. We computed the precision as the fraction of called instances which overlap ground-truth instances, and the recall as the fraction of ground-truth instances which overlap called instances. We also computed recall curves in a top-$k$ fashion, where we ranked the called instances (described above) and for each top $k$ called instances, we computed the recall relative to ground-truth instances.

For simulated datasets, the ground-truth motif instances are completely known, as they are defined at the time of sequence generation.

For experimental datasets, we obtained "ground-truth" motif instances from Vierstra et al. (2020), an independently collected set of DNA-binding footprints in various cell types. For each cell type of interest (e.g., HepG2), we simply pooled together the footprints of all experiments from that cell type and used those as a set of ground-truth motif instances.

Due to computational efficiency, we also limited the set of called motif instances for experimental datasets to the top 10000 hits.

### Evaluating QTL prioritization

We downloaded causal and non-causal QTLs in the GM12878 cell type from Lee et al. (2015). We limited the set of dsQTLs to only those in the test set of our models (i.e., on chr1). We then took our models trained on GM12878 DNase accessibility, and for each putative QTL (causal or non-causal), we computed the absolute difference in the output prediction, and treated that as a score to compute precision and auROC.

### Computing GC-content robustness

To test the robustness of our models against changes in GC content, we examined our models trained on simulated datasets (which allow us to modify the background GC content freely while keeping the motifs the same). Without retraining or fine-tuning, we evaluated the models' predictive performance on background GC content of 5%, 10%, 15%, ..., 90%, 95%.

### Generating sequences using Ledidi

To generate novel sequences using Ledidi, we took our models (ARGMINN, standard CNN, ExplaiNN) trained on our simulated SPI1 dataset. For each model, we fitted Ledidi 32 times, and for each fitted instance, we generated 32 sequences. This yielded 1024 Ledidi-generated SPI1 sequences for each model. We used $\tau = 5, \lambda = 5000$ for Ledidi.

To evaluate the quality of the generated sequences, we scanned the true PFM across each sequence (note that these models were all trained on sequences which contain motifs sampled from this exact PFM), and computed the top match score (as a cross-correlation score) for each sequence.

We computed significance of the difference in distributions (between different models used with Ledidi) of top SPI1 match scores using a one-sided Mann-Whitney U test.

### Generating adversarial examples

We generated adversarial examples in a standard CNN in two ways.

First, we generated sequences which had no motifs present, but were still predicted to have a positive label by the CNN. To do this, we built up random sequences by taking highly-activating sequences from random filters and adding them to the sequence. We took the most highly-activating sequence for each filter and randomly strung together a random ordering of such sequences to obtain a 500 bp example. We verified that no part of these sequences matched any motif (or reverse complement) in the dataset, using the same criteria as the data loader as described above.

Next, we generated sequences which had motifs, but were still predicted to be negative by the CNN. To do this, we first sampled a motif configuration from the normal simulation. Under normal circumstances, such a configuration would endow a positive label in the dataset. We then surrounded either side of this configuration with sequences that are least activating for a random ordering of filters. We randomly strung together a random ordering of such sequences to pad out a 500 bp example.

**Interpretable sequence design**

To design novel sequences intrepetably using ARGMINN, we started with our ARGMINN model trained on HepG2 DNase accessibility. We extracted motifs from the filters as described above. Prior to designing, we trimmed and filtered the motifs. Specifically, we first cut off flanks with information content lower than 0.2. We then required that after trimming, the motif was at least 5 bp and had an average information content of 0.8. We used a uniform background for computing information content.

We then extracted the motif syntax as described above, again only using the test set. This procedure takes each input sequence and assigns it a motif pattern (e.g., "CTCF - TEAD" or "FOXA" or "CTCF - FOXA - FOXA"). The motif pattern denotes which motifs were identified by ARGMINN in that sequence (in order) (e.g., CTCF followed by TEAD). Over the entire test set, we labeled each sequence with its motif pattern. For each pattern which has more than one motif, ARGMINN then gives a set of spacings that were identified over all sequences with that pattern.

We took the top 20 patterns (by number of sequences which have that pattern, requiring at least 5 sequences with that pattern). For each of the top 20 patterns, we generated 100 novel sequences with that pattern. To construct a novel sequence for a pattern, we started with an original 500 bp test sequence which falls under that pattern (sampled randomly). We then dinucleotide-shuffled the central 300 bp to destroy any functional motifs. We then inserted the motifs of that pattern (sampling from the ARGMINN-learned PFM(s)), at spacings sampled from the empirical distribution for that pattern (for single-motif patterns, the motif was simply placed in the center).

To validate our generated sequences *in silico*, we used several independently trained models as oracles:

1. Interpretations from ARGMINN itself were used to generate these sequences, but we also used ARGMINN as a predictive oracle, as it predicts accessibility from DNA sequence.

2. We used Borzoi (Linder et al., 2023) as a predictive oracle. Specifically, we used output task 1510 (trained on ENCFF577SOF), which also measured HepG2 DNase accessibility. Whereas we designed 500 bp sequences, Borzoi predicts entire signal-profile tracks from an extremely large context (524000 bp inputs). To evaluate our 500 bp sequences, we took the original coordinate from which the sequence arose, and padded it with the appropriate genomic context on either side to from the full 524 kb input. To produce a scalar prediction, we summed the center of the (binned) profile prediction corresponding to the central 500 bp input.

3. We fine-tuned Enformer (Žiga Avsec et al., 2021a) on HepG2 DNA accessibility measured by massively parallel reporter assays (MPRAs). We downloaded the MPRA data from Gosai et al. (2023), and verified that the predictive performance was on-par with the performance recorded in Gosai et al. (2023). The fine-tuning was performed using the gReLU software package.

4. We fine-tuned Enformer using gReLU, using binary labels of HepG2 DNase accessibility from a different ENCODE expriment (ENCSR291GJU) from the one ARGMINN was trained on.

We computed significance of the difference in predicted accessibility (e.g. generated vs. background, or generated vs. natural) using a one-sided Wilcoxon test.

**Expressivity requirement for REST binding**

In order to demonstrate the expressivity and interpretability of ARGMINN compared to ExplaiNN, we constructed a simulated REST dataset which explicitly tests the requirements of the unique binding syntax of REST. In this dataset, positive sequences always had the left and right motifs (in that order) with 10 bp in between. Negative sequences were structured as follows: 10% have only the left motif; 10% have only the right motif; 50% have both the left and right motifs but in improper order or orientation (e.g., the reverse-complement of left and right in that order, or the left with the reverse of the right, etc.), with each configuration having a 10 bp spacer and being selected uniformly at random; 30% have two left or two right motifs, of all sorts of configurations (e.g., forward or reverse), each having a 10 bp spacer and being selected uniformly at random; 10% are uniform background.

**Filter regularization on standard CNN**

We applied our filter regularization to a standard CNN architecture, using the same loss weights and annealing schedule as described above. We assigned identities to these motifs also using the procedure with TOMTOM as described above.

**ARGMINN loss-weight robustness**

To evaluate the robustness of ARGMINN to the weights of the filter-overlap loss and the filter-L1 loss, we trained ARGMINN on the simulated SPI1 dataset 100 times, each time with a randomly selected loss weight. The filter-overlap loss weight was randomly selected between $\lambda_o \in [10^1, 10^8]$ and the filter-L1 loss weight was randomly selected between $\lambda_l \in [10^{-5}, 10^0]$, where sampling was done uniformly on a logarithmic scale. We trained each model to completion as described above.

To evaluate the motif interpretability of each model, we extracted motifs from each ARGMINN model as described above. For each model, we then we computed the average similarity of all the discovered motifs to the true SPI1 PFM (similarity was computed as the $-\log(q)$ score from TOMTOM). If a discovered motif was not similar enough to the true PFM to pass TOMTOM's thresholds, it was given a $-\log(q)$ value of 0 by default.

