# OpenReview forum: "A mechanistically interpretable neural network for regulatory genomics"
_ICLR.cc/2025/Conference — Submitted to ICLR 2025_

### Official Review · Reviewer_DLZW · 2024-10-28

**Soundness:** 2
**Presentation:** 3
**Contribution:** 2
**Rating:** 3
**Confidence:** 4

**Summary:**

The authors consider the problem of understanding the transcription factor—DNA binding code in terms of actual binding motifs in the genome. Through a mechanistically interpretable neural network design for the prediction of protein binding events, they ensure that biologically meaningful binding motifs can be learned, showing this ability on synthetic and real data.

**Strengths:**

- The proposed architecture seems novel and thought-through.
- The methodological part is well written and easy to follow.

**Weaknesses:**

- The assumptions made in this work are prohibitively restrictive:
1) recent efforts, including large-scale consortium-based benchmarking in ENCODE-DREAM challenge [1], have shown that Epigenetics (e.g., openness of chromatin) play a decisive role in the binding of transcription factors and need to be taken into account for accurate prediction [2], a versatile set of tools has hence been designed including [3] to appropriately consider such data;
2) Genomes are unique to individuals and variants in individual genomes play a major driver in diseases such as cancer, often interrupting TF binding due to their location in motifs. Hence, recent tools consider such variants for accurate and actual genome-specific binding [4].
For both, open chromatin and variant data, how would the model incorporate this essential knowledge? How does it perform compared to existing work?
- There is a plethora of deep learning-based tools for predicting TF-binding [5,6,7,8], how does the presented approach compare to them quantitatively? With generic attribution methods, such as layer-wise-relevance propagation or SmoothGrad, extraction of motif sequences that the models respond to can be similarly achieved for these deep architectures – are your motifs qualitatively and quantitatively (e.g. similarity between discovered and known motif) better?
- It is unclear why motifs need to be learned, as there exist extensive motif databases (TRANSFAC, JASPAR, ...) that were generated through rigorous (wet-lab) experiments. Why is it better to use an (inaccurate) inferred motif than detecting motif occurrence and then building a classifier based on that? The paper would benefit from a use-case where the existing databases can not help (e.g., TFs where motifs are not available, or relevant species were no database is available).
- The quality of prediction is not properly evaluated against existing methods (including non-DL models), it is hence unclear whether the interpretability is traded off for prediction quality that would harm downstream applications. A proper comparison and evaluation in that regard would improve the paper.

[1] https://www.synapse.org/Synapse:syn6131484/wiki/402037, and associated conference tracks at RECOMB/ISCB 2016

[2] J Keilwagen, S Posch, J Grau, Accurate prediction of cell type-specific transcription factor binding Genome Biology 2019

[3] F Schmidt, F Kern, MH Schulz, Integrative prediction of gene expression with chromatin accessibility and conformation data Epigenetics & Chromatin 2020

[4] R Steinhaus, PN Robinson, D Seelow, FABIAN-variant: predicting the effects of DNA variants on transcription factor binding Nucleic Acids Research 2022

[5] L Cao et al., Prediction of Transcription Factor Binding Sites Using a Combined Deep Learning Approach Frontiers in Oncology 2022

[6] N Ghosh et al., Predicting Transcription Factor Binding Sites with Deep Learning Int J Mol Sci 2024

[7] C Chen et al., DeepGRN: prediction of transcription factor binding site across cell-types using attention-based deep neural networks BMC Bioinformatics 2021

[8] GB Kim et al., DeepTFactor: A deep learning-based tool for the prediction of transcription factors PNAS 2020

**Questions:**

- How would open chromatin and variant data be incorporated into the model? How does it perform compared to existing work?
- How does the presented approach compare to existing DL-based TF binding prediction methods and established attribution methods?
- Why is it better to use an (inaccurate) inferred motif than detecting motif occurrence and then building a classifier based on that?
- How is prediction performance affected by the architecture compared to existing work (including non-DL based methods)?

---

> ### Author Response · Authors · 2024-11-19
> **Response (1)**
>
> **Summary of rebuttal**
>
> The comparisons and suggestions cited in the review cover several different fields of research, which are different from our paper. We have clarified below what our field of interest is, and what the appropriate baselines are (and in our current manuscript, we already compare to these baselines). To prevent future confusion, we have further clarified the field of interest in the manuscript.
>
> **Clarification on the field of interest**
>
> ARGMINN is a sequence-to-function model (i.e. the input is a DNA sequence, and the output is a genomics readout, such as ChIP-seq) for *de novo* motif discovery. As described in our Related Works (Section 2), there are several studies in the field of *de novo* motif discovery from neural networks [1, 2, 3, 4, 5]. In this domain, after training a sequence-to-function model, the model is then interpreted to reveal the sequence features (i.e. motifs) which drive the particular function of interest.
>
> Importantly, the goal of ARGMINN (and related works) is *not* predictive performance; after all, the experiment used for training (e.g. ChIP-seq) is already genome-wide. Instead, the goal of ARGMINN is *interpretability*, specifically revealing sequence motifs from the model after training. This is why the setup for models such as ARGMINN uses only DNA sequence as the input, and does not rely on known databases of motifs (as that would defeat the purpose of *de novo* motif discovery).
>
> The main two classes of methods which exist today for extracting motifs from trained neural networks are: 1) interpreting the first-layer convolutional filters as motifs; and 2) computing importance scores on the input sequence and clustering high-importance regions into motifs.
>
> We compared our approach (ARGMINN) to both of these methods (and more) in our work. In each of these cases, including using importance scores, we found better results using ARGMINN (Figure 2, Supplementary Figures S1 - S5, Supplementary Table S1).
>
> **Comparisons to existing classifiers and attribution methods**
>
> In this review, seven papers were suggested as points of comparison, but they generally are not applicable to this field or this problem of *de novo* motif discovery. Citation 2 and 7 are not sequence-to-function models, as they rely on additional inputs such as chromatin accessibility, gene expression values (which estimate the free concentration of a transcription factor), known motifs, and known transcription-factor functions. As such, these models would generally not be used for motif discovery, as the decisions made by these models would include many features outside of sequence motifs, thus rendering them uninterpretable/unsuitable for this task. Additionally, although our focus is not predictive performance, these models would also not be fair comparisons for predictive performance due to the extra inputs and information. Citation 3 is a work which predicts promoter - enhancer interactions, and citation 8 predicts transcription-factor properties from the protein sequence (not DNA sequence). These two works are not in the same field as ARGMINN, and thus not comparable. Citation 4 predicts the effects of variants/changes in sequences for individual binding sites, where the motif is already given. It is also not in the same field as ARGMINN, and not comparable. Citation 5 and 6 are sequence-to-function models like ARGMINN, but are solving a different goal. These two works attempt to maximize predictive performance (predicting ChIP-seq binding) from the sequence. Both of these works use sequence embeddings to improve performance, but because of this, they are much less interpretable, and generally would not be used to perform motif discovery.
>
> In our work, we opted to compare ARGMINN to the other relevant methods in this field. First, we compared ARGMINN to another recently published mechanistically interpretable architecture, ExplaiNN. Additionally, we compared ARGMINN to a typical CNN architecture which would be used to perform motif discovery (similar in construction to the suggested citations 5 and 6, but with an architecture which is far more amenable to motif discovery). Furthermore, we compared ARGMINN to motif discovery by using importance scores. We showed results using DeepLIFTShap importance scores, which is the algorithm suggested by other experts in the field [1]. We also demonstrated that other attribution algorithms do not visibly perform better at revealing motifs, including integrated gradients and *in silico* mutagenesis (Supplementary Figure S1).

---

> > ### Comment · Reviewer_DLZW · 2024-11-22
> > **Answer rebuttal part 1**
> >
> > I do understand what your goal is, as you extensively write in your paper. What I do not understand is *why* you are doing de-novo the way you do, which is what I was elucidating in my original review above:
> >
> > (1) There exist extensive, *lab-validated* motif databases for most relevant species, *including* the examples you use in your paper. So, this knowledge already exists and is "true" knowledge.
> >
> > (2) Supposing we don't have such a database, then (as layed out in the review above) current efforts to infer new motifs are focused on including other information of, e.g., chromatin state, as it has been shown that these are necessary for good motif inference (see references above).
> >
> > (3) Although prediction is not your primary interest, if you do have a network performing badly, the extracted information might not be reflecting relationships in the actual data - it might just be the wrongly learned associations. Hence, performance *is* important. Note that at no point in the review it is mentioned that you should outperform another method, but only asked for a *proper discussion* here.
> >
> > (4) As stated above, attribution methods that are in heavy use for CNNs are ignored here. SHAP treat features to some extent in isolation, ignoring the (here sequential) structure of the input space, which is why it is not widely used for CNN analysis in practice. The referenced attribution methods above, for example LRP, perform well in recovering the relevant structure detected by an intermediate neuron, or output class. The above methods to infer TF-binding, as well as LRP, are readily available to do exactly the same task as ARGMIN. Please compare.
> >
> > (5) It seems you did not properly read my review: Citation 1-4 are used to *explain that there is more than just DNA sequence to infer motifs* and I explicitly asked how you would incorporate such essential knowledge. So yes, a direct comparison at the moment does not make sense and is not what I asked for. Rather, how would you *include* such knowledge and then compare to those methods. And to clarify, [3] *uses* enhancer-promoter relations to improve TF-binding prediction, see the conclusion section of the abstract.

---

> ### Author Response · Authors · 2024-11-19
> **Response (2)**
>
> **Incorporation of personal genomes**
>
> Although this is beyond the scope of this work, there is no technical reason why personal genomes cannot be incorporated into ARGMINN (or any sequence-to-function model). However, such data is quite rare; most (publicly available) genomics experiments are in standard cell lines, and not in personal genomes. Hence, it is standard practice to use the reference genome for training [2, 3, 4, 5], and there are very few works (if any) which focus on sequence-to-function on personal genomes.
>
> **Reliance on motif databases**
>
> The problem/field of interest in our study is *de novo* motif discovery from data (e.g. ChIP-seq), not the prediction of binding from a database of known motifs. In many ways, it is methods like ARGMINN which populate these databases of motifs in the first place.
>
> It is also worth noting that motifs currently in databases such as JASPAR are not necessarily accurate for real-world tasks. Most of these motifs are generated using *in vitro* methods (e.g. HT-SELEX), which may not provide an accurate picture of *in vivo* motifs and binding (e.g. those from ChIP-seq). They are still highly useful for distinguishing and labeling *in vivo* motifs (and we rely on these database motifs as a proxy for ground truth in our work), but they are biased in their own way.
>
> Additionally, one of the benefits of ARGMINN is the ability to learn *syntax* between motifs (not just motif discovery) (Section 4.2), and this syntax is dataset-dependent, and is not found in any database currently.
>
> **Comparison of predictive performance**
>
> While the primary goal of ARGMINN is interpretability, we recognize that maintaining reasonable predictive performance is essential, as interpretability relies on a foundation of adequate performance.
>
> Hence, we show a comparison of predictive performance of ARGMINN to other methods in Supplementary Table S5. As noted in our discussion (Section 6), it is generally well known that interpretable models tend to suffer slightly in predictive performance, as they tend to base decisions on human-interpretable concepts rather than spurious signals, which can still be informative (but are not useful for the downstream task of understanding motifs). Our results here fall in line with these expectations, although ARGMINN still outperformed the relevant baselines in many cases.
>
> We did not compare the predictive performance of ARGMINN to non-deep-learning methods, as it is generally known that deep-learning (specifically CNN-based) methods are far better than others for sequence-to-function prediction. Indeed, this was one of the biggest and most obvious trends which became clear from the ENCODE-DREAM challenge.
>
> ***Having clarified the differences between ARGMINN’s task compared to other DNA-related neural networks, as well as having demonstrated the suitability of the benchmarks used in this study for the task at hand, we respectfully request a re-evaluation of the review. We remain available to address any further questions or provide additional clarifications as needed. Thank you.***
>
> **References**
>
> [1] Shrikumar, et. al., Learning important features through propagating activation differences, ICML, 2017
>
> [2] Avsec, et. al., Base-resolution models of transcription-factor binding reveal soft motif syntax, Nature Genetics, 2021.
>
> [3] Nair, et. al, Integrating regulatory DNA sequence and gene expression to predict genome-wide chromatin accessibility across cellular contexts, Bioinformatics, 2019
>
> [4] Quang & Xie, DanQ: a hybrid convolutional and recurrent deep neural network for quantifying the function of DNA sequences, Nucleic Acids Research, 2016
>
> [5] Zhou & Troyanskaya, Predicting effects of noncoding variants with deep learning–based sequence model, Nature Methods, 2015

---

> > ### Comment · Reviewer_DLZW · 2024-11-22
> > **Answer rebuttal part 2**
> >
> > *Individual genomes*
> >
> > The review did not ask to include an entire individual's genome, but *variants* as they are a key point of interruption of TF binding in diseases such as cancer. Hence, including such information about an individual would give actual insight into TF-binding beyond what is available in the database. Such variant calls are available for most cancer datasets and *are part of clinical practice*.
> >
> > See part 1 for motif database and predictive performance discussion.
> >
> > I will keep my original rating.

---

> > > ### Author Response · Authors · 2024-12-03
> > > **Response (1)**
> > >
> > > It seems that there are still some misconceptions about the field, including its motivations and definitions. We will address each point below and try to provide additional clarification and explanation, including some resources which may be useful for learning more.
> > >
> > > **Why AI-driven *de novo* motif discovery is actually useful**
> > >
> > > AI-driven motif discovery is an entire field of research, with many scientists who are all interested in *de novo* motif discovery from data using deep learning. The field as a whole is of course aware of motif databases, but it is also well known that these motif databases are not always reliable or perfectly “true”, as suggested by the reviewer.
> > >
> > > There are several reasons why we cannot simply rely on existing motif databases. For example:
> > >
> > > 1. *in vivo* vs *in vitro* differences. As mentioned in our initial response, motifs in databases are typically measured experimentally using *in vitro* methods such as HT-SELEX. “*in vitro”* means that the measurement is not taking place from a cell, but in isolated DNA sequences. In contrast, *de novo* motif discovery using AI tends to identify motifs from *in vivo* measurements. *“in vivo”* means that the measurement is taking place in an actual cell. The differences in *in vitro* and *in vivo* measurements can be stark, and so databases are generally more inaccurate than *in vivo* measurements.
> > > 2. Sequence context is often important for motifs. TFs do not bind in pure isolation, but they depend on surrounding sequence context. Individual motifs from databases only represent the binding site, and do not capture any sequence context, including nearby flanking regions which can have a large effect on binding. Neural networks ingest an entire DNA sequence, and therefore can make decisions and learn motifs based on that sequence context, and this is something that several works (including ours) have shown (Figure 3c).
> > > 3. Motifs can depend on cellular context, including cell type and cell state (e.g., a motif in muscle cells can be a bit different from a motif for the same TF but in neurons). These cellular differences are not captured by databases, yet these subtle differences can be revealed by AI-driven *de novo* motif discovery on the appropriate cell-type-specific readout.
> > > 4. Motif syntax is missing from databases. The discovery of motif syntax (i.e., their combinations and grammatical rules) is also crucial for understanding regulatory function. Because databases treat motifs in isolation, there is no information regarding syntax. In contrast, AI methods like ARGMINN can discover and reveal this syntax.
> > >
> > > Because of these limitations (and more), motif databases are not actually perfect ground truth in all cases (even if they are very useful as a proxy). And this is one of the major reasons why the field of AI-driven motif discovery exists. For further information on this field, an excellent first step may be to consider this review paper: [1].
> > >
> > > **Where discussion of performance can be found in the manuscript**
> > >
> > > A discussion of performance is in the manuscript in Section 6 and Supplementary Table S5.
> > >
> > > **Why LRP would not be appropriate (and why DeepSHAP is still the most appropriate benchmark method)**
> > >
> > > In the above response, the reviewer suggested LRP (layer-wise relevance propagation) as a benchmark. LRP, however, is actually just another form of DeepSHAP, but with a reference of all 0s [2, 3]. It has already been shown that for DNA sequences, a reference of all 0s leads to worse explainability [2].
> > >
> > > In this field, DeepSHAP (with dinucleotide-shuffled references) remains the most common attribution framework to extract importance from neural networks for motif discovery [1, 2]. In our work, we additionally performed a brief exploration comparing DeepSHAP to *in silico* mutagenesis and integrated gradients, and found that DeepSHAP gave the best results (Supplementary Figure S1).

---

> ### Author Response · Authors · 2024-12-03
> **Response (2)**
>
> **Why the suggested citations in the original review are not relevant to this work or field**
>
> Our apologies, we are confused by this point in the review: “So yes, a direct comparison at the moment does not make sense and is not what I asked for. Rather, how would you include such knowledge and then compare to those methods.”
>
> We agree that comparing to these works does not make sense (because the goal of these works is drastically different from ARGMINN). As such, we are not sure how to think about comparing to these methods meaningfully.
>
> We’d also like to note that Citations 3 and 4 use known motifs as an input, whereas ARGMINN discovers motifs as a goal, so they are not comparable, even hypothetically.
>
> Additionally, Citation 3 actually does not use promoter - enhancer interactions to improve TF binding prediction. Instead, it’s the other way around: it uses known motifs and putative TF binding sites in order to improve the prediction of promoter - enhancer interactions. The paper does, however, *interpret* these interactions with respect to the input binding sites, which we realize can be confusing.
>
> **Why individual genomes/variants are not included in this work**
>
> We would like to clarify that individual genomes and variants are effectively synonymous: an individual’s genome is generally stored as a set of variants relative to a reference genome to save space.
>
> As noted in our initial response, in order to incorporate individual genomes/variants, we would need genome-regulatory measurements of specific cell types and specific TFs (e.g. ChIP-seq in lung cells) for many unique individuals. Although measurements on individuals is common in clinical applications (e.g., cancer studies), that is not really relevant to the field of genome regulation, where such data is unfortunately not readily available.
>
> Instead, it is effectively universal that works in our field (AI-driven motif discovery) use the reference genome in many cell lines (which can be seen in the ENCODE project). Fortunately, there is strong evidence that sequence variation across different binding sites in the single reference is sufficient for models to learn unique binding modes, without relying on individual genomes/variants [4].
>
> ***We sincerely appreciate the time taken by the reviewer to discuss. We hope our responses have provided valuable insights into the field and the context of our work. While we recognize that the critiques offered may not have been applicable to our research or the broader field, we hope that the clarifications across our responses have been sufficiently informative to raise the score from a 3.***
>
> **References**
>
> [1] Novakovsky, et. al., Obtaining genetics insights from deep learning via explainable artificial intelligence, Nature Reviews Genetics, 2022
>
> [2] Shrikumar, et. al., Learning important features through propagating activation differences, ICML, 2017
>
> [3] Lundberg and Lee, A Unified Approach to Interpreting Model Predictions, NeurIPS, 2017
>
> [4] Avsec, et. al., Base-resolution models of transcription-factor binding reveal soft motif syntax, Nature Genetics, 2021

---

### Official Review · Reviewer_fKB3 · 2024-11-02

**Soundness:** 2
**Presentation:** 2
**Contribution:** 1
**Rating:** 3
**Confidence:** 4

**Summary:**

This paper presents a neural network model aimed at providing mechanistic interpretability within the field of regulatory genomics. While the topic is relevant and important, the work suffers from several significant shortcomings that undermine its contributions and overall impact.

**Strengths:**

- Relevant Research Area: The focus on regulatory genomics is timely and addresses a critical need for interpretable models in biological applications.
- Potential Applications: The proposed methods may have applications in understanding gene regulation mechanisms, which is valuable for both computational biology and medicine.

**Weaknesses:**

- The approach lacks sufficient novelty compared to existing methods in the field, e.g., DNABERT, NT, HyenaDNA, EVO, et al. The authors do not convincingly demonstrate how their work significantly advances the state of the art in mechanistic interpretability or regulatory genomics.

- The experimental validation is inadequate. The authors fail to perform comprehensive comparisons with baseline methods or other relevant approaches. Without robust benchmarks, it isn't easy to ascertain the practical utility of the proposed model.

-  The conclusions drawn from the results are often overstated and not sufficiently supported by the data. The authors claim significant interpretability and performance improvements without adequate evidence to back these assertions.

**Questions:**

- What specific advancements does this work provide over existing models in mechanistic interpretability and regulatory genomics?
- Can the authors elaborate on their validation methodology and why they chose not to compare with established baseline methods?

---

> ### Author Response · Authors · 2024-11-19
> **Response (1)**
>
> **Summary of rebuttal**
>
> The comparisons suggested in the review are for a different (albeit related) field of research. We have clarified below what the field of interest is, and what the appropriate baselines are (and in our current manuscript, we already compare to these baselines). To prevent future confusion, we have further clarified the field of interest in the manuscript.
>
> **Clarification on the field of interest**
>
> The problem that ARGMINN is solving is the discovery of sequence motifs from a *sequence-to-function* model (i.e. the input is a DNA sequence, and the output is a functional genomics readout, such as ChIP-seq). This is very different from DNA-sequence large-language models like DNABERT and HyenaDNA (even though both models use DNA sequences as an input).
>
> Sequence-to-function models like ARGMINN are trained to predict a functional label (e.g. protein binding) from DNA sequence. In order to perform this mapping, these models implicitly learn the short sequence features (i.e. motifs) which impart function, as well as the syntax/grammar between them. These models are then subsequently *interpreted* to reveal the sequence motifs which drive function. Examples of such models include ChromDragoNN, DeepSea, DanQ, BPNet, etc., which are all slight variations on a standard CNN architecture [1, 2, 3, 4].
>
> In contrast, models like DNABERT and HyenaDNA are solving a very different task. Instead of learning a sequence-to-function map, they are unsupervised models on DNA (they do not include any functional data). Instead, they learn patterns between parts of DNA without any functional information. As such, these LLMs are generally not used to extract motifs. Hence, they are not an appropriate comparison.
>
> **Comparisons with relevant baselines, and validation methodology**
>
> In the current literature, extracting motifs from these sequence-to-function models is largely done in one of two ways. In both approaches, we start by training a convolution-based sequence-to-function model (e.g. an architecture like ChromDragoNN). Then, we apply one of:
>
> 1. Once the model is trained, motifs are extracted from the first-layer convolutional filters [3, 4, 5].
> 2. Once the model is trained, importance scores are computed for the inputs (e.g. integrated gradients), and then segmented and clustered into motifs [1, 6].
>
> In our paper, we show that both of these approaches have limited success on typical architectures. Instead, we proposed ARGMINN, a new mechanistically interpretable sequence-to-function architecture which would allow for motifs to be cleanly and successfully extracted from convolutional filters. We compared motifs discovered by ARGMINN to those discovered using these two methods, as these are the most appropriate benchmarks. We also compared our method to another model (ExplaiNN), which is an additional mechanistically interpretable architecture (of more limited expressivity and interpretability) recently published for motif discovery.
>
> We quantified the quality of motifs and motif instances in several different ways, including:
>
> - Number of unique, relevant motifs discovered by each method
> - Similarity of discovered motifs to ground-truth motifs
> - Amount of redundancy in discovered motifs
> - Number of extraneous (”garbage”) motifs discovered
> - Accuracy of motif instances in DNA sequences compared to ground truth
>
> We found that in general, ARGMINN’s interpretations were better than those discovered by the other benchmark methods, in all of these categories (Figure 2 - 3, Supplementary Figures S2 - S6, Supplementary Tables S1 - S4).

---

> > ### Author Response · Authors · 2024-11-19
> > **Response (2)**
> >
> > **Benefits relative to existing models in mechanistic interpretability and regulatory genomics**
> >
> > In the current literature, almost all motif discovery from neural networks uses one of the two methods listed above. However, another mechanistically interpretable architecture for this task, ExplaiNN, was also recently published.
> >
> > As such, in our paper, we also heavily benchmarked against ExplaiNN (Figure 2, Supplementary Tables S1 - S4). We additionally provided theoretical justification/proofs showing that ExplaiNN is less expressive than ARGMINN (Corollary 1.1).
> >
> > ***Having clarified the differences between ARGMINN’s task and DNA LLMs, as well as having demonstrated the suitability of the benchmarks used in this study for the task at hand, we respectfully request a re-evaluation of the review. We remain available to address any further questions or provide additional clarifications as needed. Thank you.***
> >
> > **References**
> >
> > [1] Avsec, et. al., Base-resolution models of transcription-factor binding reveal soft motif syntax, Nature Genetics, 2021.
> >
> > [2] Nair, et. al, Integrating regulatory DNA sequence and gene expression to predict genome-wide chromatin accessibility across cellular contexts, Bioinformatics, 2019
> >
> > [3] Quang & Xie, DanQ: a hybrid convolutional and recurrent deep neural network for quantifying the function of DNA sequences, Nucleic Acids Research, 2016
> >
> > [4] Zhou & Troyanskaya, Predicting effects of noncoding variants with deep learning–based sequence model, Nature Methods, 2015
> >
> > [5] Kelley, et. al., Basset: learning the regulatory code of the accessible genome with deep convolutional neural networks, Genome Research, 2016
> >
> > [6] Shrikumar, et. al, Technical Note on Transcription Factor Motif Discovery from Importance Scores (TF-MoDISco) version 0.5.6.5, arXiv, 2018

---

> ### Author Response · Authors · 2024-11-27
>
> To the reviewer,
>
> We have clarified the main objectives and contributions of our work both in the manuscript and in our response. With these core aspects now clarified, and as the end of the discussion period is approaching, we would like to ask if there are any remaining concerns upon re-evaluation of our work. We are happy to answer any further questions and confusions. Thank you.
>
> -Authors

---

### Official Review · Reviewer_RgNo · 2024-11-02

**Soundness:** 3
**Presentation:** 2
**Contribution:** 2
**Rating:** 6
**Confidence:** 2

**Summary:**

This paper designed a novel mechanistically interpretable architecture for regulatory genomics, where motifs and their syntax are directly encoded and readable from the learned weights and activations. Through several experiments, the authors show that ARGMINN excels in
de novo motif discovery and motif instance calling, is robust to variable sequence contexts, and enables fully interpretable generation of novel functional sequences

**Strengths:**

1. This paper successfully applied the MI methods to the regulatory genomics domain
2. This paper provided thorough assessment for the usage of ARGMINN
3. The authors provided detailed related works and theoretical results based on first-order logic

**Weaknesses:**

1. I find this work may be not interesting to the most readers of ICLR. This paper should fall into a biological journal (e.g. journals for stastical geomics)
2. The technologies introduced in this paper are actually not based on biological mechanics but common mechanical Interpretation tricks including regularizations, separate convolutions and attention for interpretability.

**Questions:**

1. Could you please explain more about the connection between ARGMINN and CBMs?
2. Could you please explain more about the difference between ARGMINN and ExplaiNN?

---

> ### Author Response · Authors · 2024-11-19
> **Response (1)**
>
> **Relevance to ICLR**
>
> We believe that such ML research for biology certainly does have a place in conferences such as ICLR. Indeed, this exact problem of extracting biological sequence motifs from DNA models is the subject of DeepLIFT (published in ICML), an algorithm which uses importance scores to reveal putative DNA motifs from a trained model (and we compare to this algorithm in our work) [1]. It was also very recently that HyenaDNA (a similar work which developed an architecture for a different task on DNA sequences) earned a spotlight at NeurIPS. [2]. ICLR this year also featured multiple papers on DNA-sequence models and their biological applications [3, 4].
>
> Although the motivating problem and application are biological in nature, our work is still one of the early few which explores mechanistic interpretability at such a deep level [5]. In our work, we designed a novel architecture using fundamental ML techniques, but tailored these techniques to the domain at hand. In many ways, these domain-informed constraints are required to construct a mechanistically interpretable architecture which is both interpretable and sufficiently expressive [5]. In our paper, we additionally gave mathematical/theoretical justification for our architecture and these design decisions, as well as empirical results supporting their benefits.
>
> **Biophysical vs mechanistically interpretable**
>
> To clarify, ARGMINN is not meant to be a *biophysical* model. That is, although parts of the architecture (e.g. the decision to learn motifs as convolutional filters, which is ubiquitous in this area) are inspired by biophysics, the model itself is not intended to encode biophysical processes (which would take far too much processing power to represent accurately, particularly the interactions between motifs).
>
> There is a need to learn motifs and their interactions in a data-driven manner, but currently, it is difficult to extract/interpret the learned motifs and their interactions from trained models. In our work, we take the approach of constructing a mechanistically interpretable architecture, where the learned decisions are based on human-interpretable motifs. The techniques we apply are intentionally those which help a model be more mechanistically interpretable, thereby revealing the learned decision rules [5].
>
> Indeed, we show that ARGMINN achieves far better results in motif discovery with its mechanistically interpretable architecture compared to the state-of-the-art methods today, even though it is not a biophysical model.
>
> **ARGMINN vs CBMs**
>
> CBMs take an input (e.g. an image), and map it to a set of higher-order, human-interpretable concepts (e.g. for a cat image, these features may be binary labels of “whisker”, “pointy ear”, “stripes”, “tail”). These concepts are then passed to another classifier which produces a final output by implicitly learning the logic between these features (e.g. “whisker” + “pointy ear” + “tail” = cat).
>
> ARGMINN can be viewed as a kind of CBM, where the input (a DNA sequence) is mapped to a set of higher-order, human-interpretable concepts (motifs). The attention layers in ARGMINN then learn the final output by learning the logic between these motifs.
>
> ARGMINN differs from typical CBMs in two main ways:
>
> - In a typical CBM, the only part of the model which is interpretable is the concepts (e.g. “whisker”). But in ARGMINN, the attention layers are designed such that the *combination* of the motifs/concepts is also interpretable. That is, we can easily identify which concepts are being combined to produce the final output. This is not typically achievable (easily) in a standard CBM.
> - CBMs typically require labeled concepts for each input (e.g. to train a CBM on images, we need annotations for which images have whiskers or stripes). ARGMINN, however, does not require concept/motif labels at all. Owing to the filter regularization we developed, it learns all of the relevant motifs (i.e., “concepts”) from scratch, and separates them out into individual channels.
>
> **ARGMINN vs ExplaiNN**
>
> ARGMINN and ExplaiNN both attempt to solve a similar problem: train a model predicting biological function from DNA sequences, and subsequently interpret sequence motifs from the trained model.
>
> ExplaiNN’s architecture learns a set of single-filter CNN modules, where each CNN module learns one motif (in its one filter), and produces a scalar value for an input DNA sequence. ExplaiNN’s final output is a learned linear combination of these scalar values.
>
> In contrast with ARGMINN, ExplaiNN does not have any regularization to prevent multiple CNN modules from learning the same motif; information is still distributed across the network which makes it less interpretable (Figure 2). Additionally, ExplaiNN is limited by a linear combination (instead of a multi-layer attention mechanism), and so ExplaiNN is also limited in its expressivity (Corollary 1.1).

---

> > ### Author Response · Authors · 2024-11-19
> > **Response (2)**
> >
> > **References**
> >
> > [1] Shrikumar, et. al., Learning important features through propagating activation differences, ICML, 2017
> >
> > [2] Nguyen, et. al., HyenaDNA: Long-Range Genomic Sequence Modeling at Single Nucleotide Resolution, NeurIPS, 2023
> >
> > [3] Marin, et. al., BEND: Benchmarking DNA Language Models on Biologically Meaningful Tasks, ICLR, 2024
> >
> > [4] Zhou, et. al.,  DNABERT-2: Efficient Foundation Model and Benchmark For Multi-Species Genomes, ICLR, 2024
> >
> > [5] Bereska & Gavves, Mechanistic Interpretability for AI Safety -- A Review, arXiv, 2024

---

> > > ### Author Response · Authors · 2024-11-27
> > >
> > > To the reviewer,
> > >
> > > We have addressed the issues brought up in the review, and as the end of the discussion period is approaching, we would like to ask if there are any remaining concerns which we may address. We are happy to answer any further questions. Thank you.
> > >
> > > -Authors

---

> > > > ### Comment · Reviewer_RgNo · 2024-12-01
> > > >
> > > > Sorry for the late reply. The applicability of AI4Genomics in top AI conferences is indeed controversial. Some argue that it is merely an issue of interpretability set against a biological background (without introducing new problems or innovations in AI), and should be prioritized for submission to biological journals. Others believe it is a legitimate application of AI to genomics. Although I support the former view, I remain open to the authors' perspective. Personally, I think it is acceptable whether the paper is accepted or not, and I will raise my score to 6.  I wish the authors good luck.

---

### Official Review · Reviewer_EbEY · 2024-11-03

**Soundness:** 2
**Presentation:** 2
**Contribution:** 2
**Rating:** 5
**Confidence:** 3

**Summary:**

This paper proposes a mechanistically interpretable neural network to solve several genome understanding tasks, including regulatory function prediction, DNA motif discovery and motif interaction prediction. The proposed architecture consists of a motif-scanner, a regularizer and a syntax builder to respectively recognize motifs, filter out wrong patterns and predict interactions between motifs. The paper did many analyses to show the discovered motifs, motif instance calling, QTL prioritization and the architecture robustness.

**Strengths:**

The paper is targeting at an important task, and shows many cases for each evaluated task, which is straightforward.

**Weaknesses:**

1. The paper is hard to follow. After reading the manuscript, I know the authors develop a new architecture consisting of several components, and each of them achieves a specific function. However, I cannot figure out the detailed contributions of this paper. That is to say, what are the differences of the proposed architecture compared to previous method?

2. The experimental settings and results are not complete.:

2.1 The paper didn't show any specific number on the evaluated tasks. All results are shown in Figure. Though we could see some improvements compared to baselines, we still don't know what specific performance the proposed model can achieve.

2.2 The paper lacks ablation study.

2.3 The paper lacks specific experimental settings. For example, what are the specific values for n_f and n_L?  Do the standard CNN have the same layer as ARGMINN?

2.4 What species genome sequences are used for each task?

**Questions:**

The most uncertain parts for me are the contributions and the detailed experimental settings of the proposed method.

---

> ### Author Response · Authors · 2024-11-19
> **Response (1)**
>
> **Summary of rebuttal**
>
> We have clarified the novelty of our method compared to previous works below. All other requests in the review (e.g. specific numbers, ablation studies, experimental settings, etc.) are in the supplement of the manuscript (and referenced from the main text). To prevent future confusion, we have further clarified the background of the field in the manuscript.
>
> **Differences of proposed architecture to previous methods**
>
> ARGMINN is a sequence-to-function model: the input is a DNA sequence, and the output is a functional genomics readout, such as protein binding. These sequence-to-function models (like ARGMINN) are trained to predict this functional readout (e.g. protein binding) from the input DNA sequence. In order to perform this mapping, these models must implicitly learn the short sequence features (i.e. motifs) which define function, as well as the syntax/grammar between them. The chief goal of these models is not predictive performance, but subsequent *interpretation*. After training, these models are then *interpreted* to reveal the sequence motifs which drive function. Examples of such models include ChromDragoNN, DeepSea, DanQ, BPNet, etc., which are all slight variations on a standard CNN architecture [1, 2, 3, 4].
>
> In the current literature, extracting motifs from these sequence-to-function models is largely done in one of two ways. In both approaches, we start by training a convolution-based sequence-to-function model (e.g. an architecture like ChromDragoNN). Then, we apply one of:
>
> 1. Once the model is trained, motifs are extracted from the first-layer convolutional filters [3, 4, 5].
> 2. Once the model is trained, importance scores are computed for the inputs (e.g. integrated gradients), and then segmented and clustered into motifs [1, 6].
>
> In our paper, we show that both of these approaches have limited success on typical architectures. Instead, we proposed ARGMINN, a new mechanistically interpretable sequence-to-function architecture which would allow for motifs to be cleanly and successfully extracted from convolutional filters. We compared motifs discovered by ARGMINN to those discovered using these two methods, as these are the most appropriate benchmarks. We also compared our method to another model (ExplaiNN), which is an additional mechanistically interpretable architecture (of more limited expressivity and interpretability) recently published for motif discovery.
>
> In order to allow for motifs to be successfully extracted from convolutional filters, our mechanistically interpretable architecture introduces two main novelties: the convolutional layer with a filter-overlap regularizer (Section 3.1), and an attention mechanism which explicitly learns motif interactions (Section 3.2).
>
> In the current literature, the standard architectures (e.g. ChromDragoNN, DanQ, etc.) consist of one or more convolutional layers (without any special regularization), followed by additional layers to learn interactions (e.g. fully-connected layers, dilated convolutional layers, standard transformers, etc.) [1, 2, 3, 4].
>
> Compared with previous methods, our approach introduces a filter-overlap regularizer which makes the first-layer convolutional layer interpretable and reveals motifs directly. Additionally, our custom attention mechanism contrasts with existing works, as it allows for long-range interactions while remaining interpretable, which allows it to reveal motif syntax (an ability which previous architectures did not have at all).
>
> In our work, we also included both theoretical and experimental results showing the improved interpretability and expressivity of ARGMINN relative to ExplaiNN.

---

> ### Author Response · Authors · 2024-11-19
> **Response (2)**
>
> **Numbers for evaluated tasks**
>
> Figure 2 shows both comprehensive and examples of quantitative results comparing ARGMINN to several benchmark methods. The same data but in tabular form can be found in Supplementary Tables S1 - S4.
>
> **Ablation studies**
>
> We showcased an ablation study (Section 6, L522) in which we applied our filter regularization to a standard CNN. We showed that with our filter regularization, we were able to make the standard CNN far more interpretable and also reveal high-quality motifs (Supplementary Figure S10).
>
> This ablation shows the marginal effects of the filter-overlap regularization, the first of our two architectural novelties. Note that it would not be meaningful to ablate the second novelty (i.e. the modified attention layers), as it is only useful for interpretability when the input activations are separated into individual motifs (which is what the filter regularization does).
>
> **Experimental details**
>
> The details for the experimental settings can be found in the supplementary methods. All values of hyperparameters, including $n_{f}$ and $n_{L}$, are in Supplementary Table S7.
>
> **Species**
>
> All of our experiments are on human genomes and human motifs.
>
> ***Having clarified ARGMINN’s task and its novelties and contributions, as well as having pointed out the experimental methodologies, we respectfully request a re-evaluation of the review. We remain available to address any further questions or provide additional clarifications as needed. Thank you.***
>
> **References**
>
> [1] Avsec, et. al., Base-resolution models of transcription-factor binding reveal soft motif syntax, Nature Genetics, 2021.
>
> [2] Nair, et. al, Integrating regulatory DNA sequence and gene expression to predict genome-wide chromatin accessibility across cellular contexts, Bioinformatics, 2019
>
> [3] Quang & Xie, DanQ: a hybrid convolutional and recurrent deep neural network for quantifying the function of DNA sequences, Nucleic Acids Research, 2016
>
> [4] Zhou & Troyanskaya, Predicting effects of noncoding variants with deep learning–based sequence model, Nature Methods, 2015
>
> [5] Kelley, et. al., Basset: learning the regulatory code of the accessible genome with deep convolutional neural networks, Genome Research, 2016
>
> [6] Shrikumar, et. al, Technical Note on Transcription Factor Motif Discovery from Importance Scores (TF-MoDISco) version 0.5.6.5, arXiv, 2018

---

> > ### Author Response · Authors · 2024-11-27
> >
> > To the reviewer,
> >
> > We have clarified the main objectives and contributions of our work both in the manuscript and in our response. With these core aspects now clarified, and as the end of the discussion period is approaching, we would like to ask if there are any remaining concerns upon re-evaluation of our work. We are happy to answer any further questions and confusions. Thank you.
> >
> > -Authors

---

### Official Review · Reviewer_Zz46 · 2024-11-04

**Soundness:** 3
**Presentation:** 4
**Contribution:** 4
**Rating:** 6
**Confidence:** 3

**Summary:**

The paper introduces ARGMINN, a novel DNN architecture designed for recovering motifs with mechanistic interpretability. The model is designed to directly encode biologically meaningful motifs in its weights and activations. Exhaustive experiments show ARGMINN's strength in motif discovery, instance calling, robustness, and interpretable sequence generation. This work successfully narrows the gap between the high predictive performance of DNNs and the need for interpretability in genomics prediction.

**Strengths:**

- The proposed ARGMINN model brings the mechanistically interpretable architecture into a classical DNN architecture, trying to address the Interpretability challenge in applying deep learning to genomics.
- The authors provide theoretical proofs showing that their model can recognize all motif configurations.
- ARGMINN shows superior performance  across all tasks, compared to baselines (e.g. CNNs and ExplaiNN). And the authors also showcase the model's ability to generate novel functional sequences based on the learned motifs, which is a valuable contribution.

**Weaknesses:**

- The evaluation strategy used in the paper involves reserving chr8 and chr10 for validation and chr1 for testing, while training on all other chromosomes. However, because chromosomes can vary in genomic features, using a specific chromosome split might introduce biases in performance evaluation.
- Since the paper introduces a memory stream into the attention mechanism, it would be helpful to include a discussion on the training times and memory usage of the model.

**Questions:**

- For the evaluation strategy, why do you choose chr8 and chr10 for validation and chr1 for testing? Have you considered to use leave-one-out strategy to assess the model's generalization across different genomic regions?
- Could you provide details on the computational resources required to train ARGMINN?

---

> ### Author Response · Authors · 2024-11-19
> **Response**
>
> **Selection of chromosome splits**
>
> Using chromosome splits to define training/validation/test sets for regulatory genomics is standard practice in this area, as it completely prevents contamination of the same sequence (or part of the same sequence) from occurring in both training and testing [1, 2, 3, 4]. Chromosome splits also help prevent weaker contamination arising from genetic duplication events. That is, much of the genome arises from sections of DNA being duplicated across the same chromosome, so chromosome splits also prevent different sequences (but with a recently shared history) from occurring in both training and testing.
>
> These chromosome splits were directly adopted from prior literature in a similar space, and they were selected to obtain roughly a 80/10/10 split of training/validation/testing (the human chromosomes have a wide variety of sizes, so the selected chromosomes in each set are not consecutive). Prior work in ML on regulatory DNA has also shown that different folds of chromosomes yield similar results (the regulatory signals across the genome are fairly uniform by chromosome, likely owing to their large sizes) [1, 2].
>
> **Computational resources used**
>
> We trained all of our models on a single Nvidia A100 GPU. As for training times, here we report the training time for our SPI1 dataset (others are quite similar). Note that the increased relative training time of ARGMINN largely stems from the computation of the filter-overlap regularization loss.
>
> | **Model** | **Training time (mins)** | **Number of parameters** |
> | --- | --- | --- |
> | ARGMINN | 15 | 13000 |
> | Standard CNN | 6 | 2100 |
> | ExplaiNN | 8 | 58000 |
>
> We have added this table to the supplement of our manuscript.
>
> **References**
>
> [1] Avsec, et. al., Base-resolution models of transcription-factor binding reveal soft motif syntax, Nature Genetics, 2021.
>
> [2] Nair, et. al, Integrating regulatory DNA sequence and gene expression to predict genome-wide chromatin accessibility across cellular contexts, Bioinformatics, 2019
>
> [3] Quang & Xie, DanQ: a hybrid convolutional and recurrent deep neural network for quantifying the function of DNA sequences, Nucleic Acids Research, 2016
>
> [4] Zhou & Troyanskaya, Predicting effects of noncoding variants with deep learning–based sequence model, Nature Methods, 2015

---

> > ### Author Response · Authors · 2024-11-27
> >
> > To the reviewer,
> >
> > We have addressed the issues brought up in the review, and as the end of the discussion period is approaching, we would like to ask if there are any remaining concerns which we may address. We are happy to answer any further questions. Thank you.
> >
> > -Authors

---

### Meta-Review · Area_Chair_6ruV · 2024-12-21

**Metareview:**

**a) Scientific Claims and Findings:**
The paper introduces a novel neural network architecture, ARGMINN, designed for analyzing DNA sequences in regulatory genomics. The primary objective is to develop a model that not only predicts genomic readouts, such as protein–DNA binding, but also provides direct interpretability by revealing underlying motifs and their syntactical relationships from the learned weights and activations. Traditional methods often struggle with interpretability due to the dispersion of information across filters and layers, and reliance on unstable importance scores. In contrast, this architecture encodes motifs and their syntax in a manner that is directly readable post-training. The authors provide both theoretical and empirical evidence supporting the model's full expressivity while maintaining high interpretability. Through various experiments, the architecture demonstrates excellence in de novo motif discovery, motif instance identification, robustness to variable sequence contexts, and the ability to generate novel functional sequences with full interpretability.

**(b) Strengths:**
* Enhanced Interpretability: The architecture allows for direct reading of motifs and their syntactical rules from learned weights and activations, addressing a significant challenge in regulatory genomics.
* In contrast to CBMs, ARGMINN does not require concept/motif labels.
* Theoretical and Empirical Validation: The model's expressivity and interpretability are supported by both theoretical analysis and empirical experiments. The theory states that in comparison to the competitor explaiNN, the proposed ARGMINN architecture is more expressive.
* Robust Performance: The architecture excels in tasks such as de novo motif discovery and motif instance identification, and shows robustness to variable sequence contexts, indicating its practical applicability.

**(c) Weaknesses:**
* Scalability Concerns: The paper does not provide a detailed analysis of the model's scalability to large genomic datasets, which is crucial for real-world applications in regulatory genomics. Additional experiments during the rebuttal suggest that ARGMINN requires a relatively high number of parameters (relative to the studied CNN) and increased training time in comparison with competitors. However, runtimes were generally not significant in the reported table.
* Comparative Analysis: While the model demonstrates promising results, a more comprehensive comparison with existing state-of-the-art models would help in understanding its relative performance and potential advantages. Reviewer fKB3 requested comparisons with DNABERT, NT, HyenaDNA, EVO, for instance. As the authors argue, such models have been trained for a different purpose, but they list other models ChromDragoNN, DeepSea, DanQ, BPNe with which they still do not compare. Also Reviewer DLZW has suggested a couple of methods for comparison.
* Scope of the Approach: Reviewer DLZW has raised concerns about the impact of the task and whether novel discovery is possible. The paper would benefit from a use-case where the existing databases cannot help (e.g., TFs where motifs are not available, or relevant species were no database is available).
* Ablations: Additional requested ablations suggest that the filter-overlap regularization has only a marginal effect. It’s main purpose is to provide interpretability in combination with the modified attention layer (in the sense that motifs can be identified).

**(d) Reasons for Rejection:**
After careful consideration, the decision to reject the paper is based on the following reasons:
1. Insufficient Scalability Analysis: The absence of a thorough examination of the model's scalability to large genomic datasets raises concerns about its applicability in practical, large-scale regulatory genomics studies.
2. Limited Comparative Evaluation: The paper does not provide an extensive comparison with existing models, making it challenging to assess the proposed architecture's relative performance and contributions to the field.
3. Limited Impact (Methodological Innovation or Application Relevance):  Reviewer RgNo has criticised a limited methodological innovation, as ARGMINN combines known components and relies on established intuition. This would not be a limiting factor if the empirical evidence for application relevance was very high. Yet, novel discovery is not supported by wet lab experiments, for instance, (see Reviewer DLZW).
Addressing these concerns would enhance the paper's contribution and its potential for acceptance in future submissions.

**Additional Comments On Reviewer Discussion:**

First and foremost, based on the discussion with Reviewer RgNo, we feel the need to emphasize that applications papers in computational biology are generally welcome at ICLR and highly appreciated.
They are measured against common standards of the field regarding their application relevance and methodological contributions. As detailed in the above meta review, several reviewers suggested that the paper in its current form could improve at least regarding one dimension and requested additional experiments and/or empirical evidence that the proposes method provides merit in novel discovery beyond known motifs.

---

### Decision · Program_Chairs · 2025-01-22

Reject